# Scattering approach to diffusion quantifies axonal damage in brain injury

Ali Abdollahzadeh ®[1,2] ✉, Ricardo Coronado-Leija[1], Hong-Hsi Lee ®[3], Alejandra Sierra ®[2], Els Fieremans ®[1] & Dmitry S. Novikov ®[1] ✉

Early diagnosis and noninvasive monitoring of neurological disorders require sensitivity to elusive cellular-level alterations that occur much earlier than volumetric changes observable with the millimeter-resolution of medical imaging modalities. Morphological changes in axons, such as axonal varicosities or beadings, are observed in neurological disorders, as well as in development and aging. Here, we reveal the sensitivity of time-dependent diffusion MRI (dMRI) to the structurally disordered axonal morphology at the micrometer scale. Scattering theory uncovers the two parameters that determine the diffusive dynamics of water along axons: the average reciprocal cross-section and the variance of long-range cross-sectional fluctuations. This theoretical development allows us to predict dMRI metrics sensitive to axonal alterations over tens of thousands of axons in seconds rather than months of simulations in a male rat model of traumatic brain injury, and is corroborated with ex vivo dMRI. Our approach bridges the gap between micrometers and millimeters in resolution, offering quantitative and objective biomarkers applicable to a broad spectrum of neurological disorders.

Neurological disorders are a global public health burden, with their prevalence expected to rise as the population ages[1]. A ubiquitous signature of a wide range of these pathologies is the change in axon morphology at the micrometer scale. Such changes are extensively documented in Alzheimer's[2,3], Parkinson's[4,5], and Huntington's[6,7] diseases, multiple sclerosis[8–10], stroke[11–13], and traumatic brain injury (TBI)[14–16]; they are also implicated in development[17,18] and aging[19,20]. In particular, within neurodegenerative disorders[2–7], abnormalities in the axon morphology involve disruptions in axonal transport[21–24] and the aberrant accumulation of cellular cargo[23,24] comprising mitochondria, synaptic vesicles, or membrane proteins and enzymes[22]. This buildup forms a transport jam, often presenting itself in terms of axonal varicosities or beadings[2,5,21,25], contributing to abnormal morphological changes along axons − a unifying microstructural disease hallmark, notwithstanding the wide heterogeneity of clinical symptoms.

Detecting and quantifying the key micrometer-scale changes[3,25–29] that precede macroscopic atrophy or edema are unmet clinical needs and technological challenges−given that in vivo biomedical imaging operates at a millimeter resolution. Across a spectrum of non-invasive imaging techniques, including recent advancements in ionizing radiation[30], super-resolution ultrasound[31,32] and MRI[33], diffusion MRI (dMRI) is uniquely sensitive to nominally invisible tissue microgeometry at the scale of the water diffusion length $\ell \sim 1–10\,\mu m$, $2 − 3$ orders of magnitude below the millimeter-size imaging voxels[34–37]. The diffusion length $\ell(t) \equiv \langle x^2(t) \rangle^{1/2}$ is the root mean square displacement of water molecules, which carry nuclear spins detectable via MRI; at typical diffusion times $t \sim 1–100\,ms$, it is commensurate with dimensions of cells and organelles, offering an exciting prospect for non-invasive in vivo histology at the most relevant biological scale[38–40]. Realizing the ultimate diagnostic potential of biomedical imaging hinges on our ability to interpret macroscopic measurements in terms of specific features of tissue microgeometry. This interpretation relies on biophysical modeling[37,41,42] to identify the few relevant degrees of freedom that survive the massive averaging of local tissue microenvironments of the size $\sim \ell(t)$ within a macroscopic voxel.

[1]Center for Biomedical Imaging, Department of Radiology, New York University School of Medicine, New York, NY, USA. [2]A.I. Virtanen Institute for Molecular Sciences, University of Eastern Finland, Kuopio, Finland. [3]Athinoula A. Martinos Center for Biomedical Imaging, Department of Radiology, Massachusetts General Hospital, Harvard Medical School, Boston, MA, USA. ✉e-mail: ali.abdollahzadeh@uef.fi; dmitry.novikov@nyulangone.org

## Results

Here we identify the morphological parameters associated with pathological changes in axons that can be probed with dMRI measurements − thereby establishing the link between cellular-level pathology and noninvasive imaging. Specifically, we analytically connect the axonal microgeometry (Fig. 1) to the time-dependent along-axon diffusion coefficient (Fig. 2)

$$D(t) \equiv \frac{\ell^2(t)}{2t} \simeq D_\infty + \frac{c_D}{\sqrt{t}}, \quad t \gg t_c. \tag{1}$$

As discussed below, $D(t)$ is accessible with dMRI[43–49] as the along-tract diffusion coefficient in the clinically feasible regime of diffusion time $t$ exceeding the correlation time $t_c \sim 1\,\text{ms}$ to diffuse past $\mu m$-scale axon heterogeneities.

By developing the scattering framework for diffusion in a tube with varying cross-sectional area $A(x)$ along its length $x$ (cf. *Methods* section), we derive Eq. (1) and find exact expressions for its parameters $D_\infty$ and $c_D$ in terms of the relative axon cross-section $\alpha(x)$ (Fig. 2a):

$$\frac{D_0}{D_\infty} = \left\langle \frac{1}{\alpha(x)} \right\rangle, \quad \alpha(x) = \frac{A(x)}{\bar{A}}, \tag{2}$$

where $\bar{A} = \langle A(x) \rangle$ is the mean cross-section, and

$$c_D = 2\Gamma_0 \sqrt{\frac{D_\infty}{\pi}}. \tag{3}$$

In Eq. (2), $D_0$ is the intrinsic diffusion coefficient in the axoplasm, and the geometry-induced attenuation of the diffusion coefficient $D_0/D_\infty$ (the tortuosity factor) is given by the reciprocal relative cross-section averaged along the axon. In Eq. (3), $\Gamma_0 = \Gamma_\eta(q)|_{q\to 0}$ is the small-$q$ limit of the power spectral density $\Gamma_\eta(q) = \eta(-q)\eta(q)/L$ (with the dimensions of length), where the dimensionless stochastic variable $\eta(x) = \ln \alpha(x)$ is a relevant measure of cross-sectional fluctuations $A(x)$. The limit $\Gamma_0$ is a measure of *axon shape heterogeneity at large spatial scales*, and $L \sim 100\,\mu m$ is the macroscopic length of an axon segment, Fig. 2 (cf. Eq. (19) in *Methods*).

The theory (1)–(3) distills the myriad parameters necessary to specify the geometry of irregular-shaped axons (e.g., those segmented from serial block-face scanning electron microscopy (SBEM) volumes[16,50,51], as in Fig. 1) into just two parameters in Eq. (1): the long-time asymptote $D_\infty$, and the amplitude $c_D$ of its $t^{-1/2}$ power-law approach (i.e., of the sub-diffusive correction to the growing mean-squared displacement $\ell^2(t)$). These parameters are further exactly related to the two characteristics of the stochastic axon shape variations $\alpha(x)$ along its coordinate $x$, Eqs. (2)–(3), thereby bridging the gap between millimeter-level dMRI signal and micrometer-level changes in axon morphology, expressed in forming beads or varicosities that can occur in response to a variety of pathological conditions and injuries. In what follows, we offer the physical intuition and considerations leading to the above results, validate them using Monte Carlo simulations (Fig. 2), and illustrate our findings in a TBI pathology that is particularly difficult to detect with noninvasive imaging (Figs. 3 and 4).

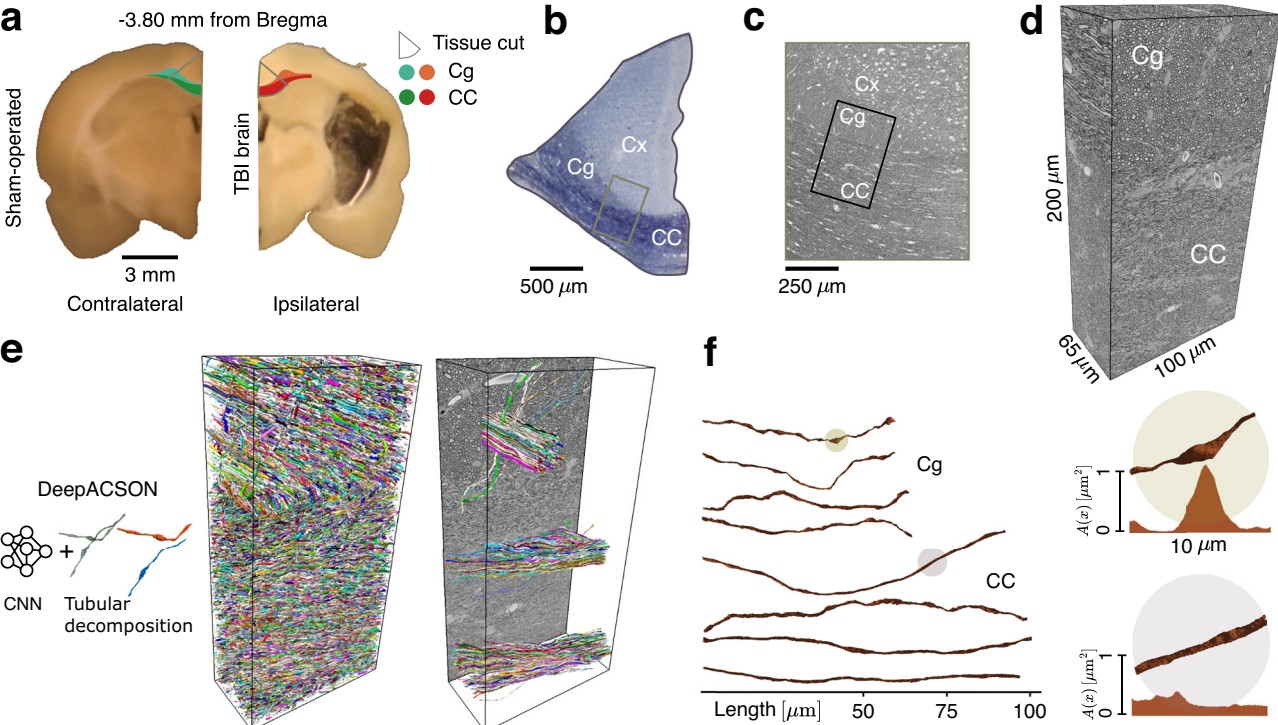

**Fig. 1 | Axon segmentation and morphology. a** Representative photomicrographs of 1 mm thick coronal sections, with the cingulum (Cg) and corpus callosum (CC) highlighted. Selected sections for staining encompass parts of the CC, Cg, and cerebral cortex (Cx). **b** A photomicrograph of a semi-thin section stained with toluidine blue, with a block trimmed further for the serial block-face scanning electron microscopy[107] (SBEM) technique. **c** A low-resolution EM image to navigate for the final SBEM imaging. **d** A representative SBEM volume, voxel size $50 \times 50 \times 50\,\text{nm}^3$, from a large field-of-view $200 \times 100 \times 65\,\mu m^3$ that retains two-thirds CC and one-third Cg. **e** DeepACSON[16,60], a convolutional neural network (CNN)-based technique (see *Methods*) segmented tens of thousands of myelinated axons in each SBEM volume; we sampled and visualized myelinated axons at three random positions. **f** Micrometer-scale along-axon shape variations of representative myelinated axons from Cg and CC. Two 10 $\mu m$-fragments of axons within the shaded circles are zoomed in: the corresponding cross-sectional areas $A(x)$ show a substantial variation (e.g., beading) in one axon and a relative uniformity in the other one. Source data are provided as a Source Data file.

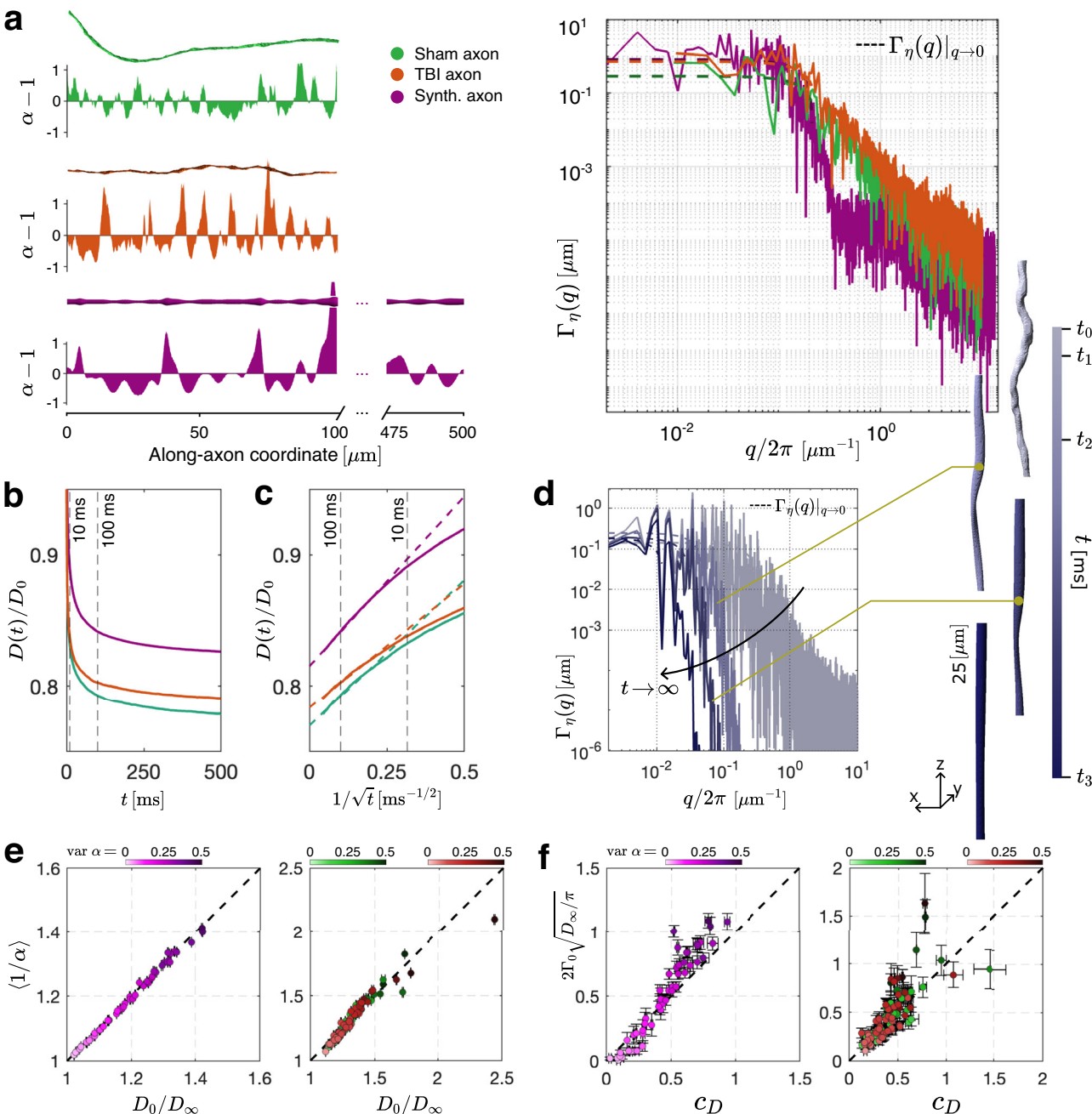

**Fig. 2 | From axon geometry to along-axon diffusivity. a** Relative cross-sectional variations $\alpha$ for representative SBEM-segmented myelinated axons (sham and TBI), the synthetic axon, and their power spectral densities $\Gamma_\eta(q)$. The finite plateau $\Gamma_0 = \Gamma_\eta(q)|_{q\to 0} > 0$ signifies the short-range disorder (finite correlation length) in the cross-sections. **b** Monte Carlo simulated $D(t)$ ensemble-averaged over $N_{\mathrm{axon}} = 50$ randomly synthesized, $N_{\mathrm{axon}} = 43$ randomly sampled SBEM myelinated sham and $N_{\mathrm{axon}} = 57$ TBI axons, with colors corresponding to (**a**). **c** $D(t)$ for all three cases scales asymptotically linearly with $1/\sqrt{t}$, validating the functional form of Eq. (1). **d** Coarse-graining over the increasing diffusion length $\ell(t)$ makes an axon appear increasingly more uniform, suppressing shape fluctuations $\Gamma_\eta(q)$ with $q \gtrsim 1/\ell(t)$, such that only the $q \to 0$ plateau $\Gamma_0$ "survives" for long $t$ and governs the diffusive dynamics (1). To illustrate the effect, an axon segment is Gaussian-filtered with the standard deviation $\ell(t_i)/\sqrt{2}$ for $\ell(t_i) = 0, 5, 10, 20\,\mu m$. The coarse-graining of the axon segment along its length is color-coded for increasing diffusion times $t_i$ according to the color bar. **e** The exact tortuosity limit (2) is validated for both synthetic and SBEM individual axons. Axons with larger cross-sectional variations var $\alpha$ have higher tortuosity. The center represents the mean, and horizontal error bars reflect errors in estimating $D_\infty$ from Eq. (1) (see *Methods*). **f** The predicted amplitude $c_D$ of the $t$-dependent contribution to $D(t)$, Eq. (3), validated against its MC counterpart estimated from Eq. (1), for individual synthetic and TBI axons (colors as in (**a**)). The coefficient $c_D$ is larger for axons with greater cross-sectional variations. The filled circles and error bars reflect means and errors in estimating $c_D$ from MC-simulated $D(t)$ (horizontal) and estimating the plateau $\Gamma_0$ from $\Gamma_\eta(q)$ (vertical), as shown by dashed lines in the power spectral densities of (**a**) (see *Methods*). The number of samples is indicated in (**b**). Source data are provided as a Source Data file.

## Physical picture and the scattering problem

The physical intuition behind the theory (1)–(3) is as follows. Averaging of the reciprocal relative cross-section in Eq. (2) is rationalized via the mapping between diffusivity $D_\infty$ and dc electrical conductivity; an axon

is akin to a set of random elementary resistors with resistivities $\sim 1/\alpha(x)$, and resistances in series add up (see *Methods*). The qualitative picture for Eq. (3) involves realizing that an axon, as effectively "seen" by diffusing water molecules, is *coarse-grained*[37,40,52] over an increasing

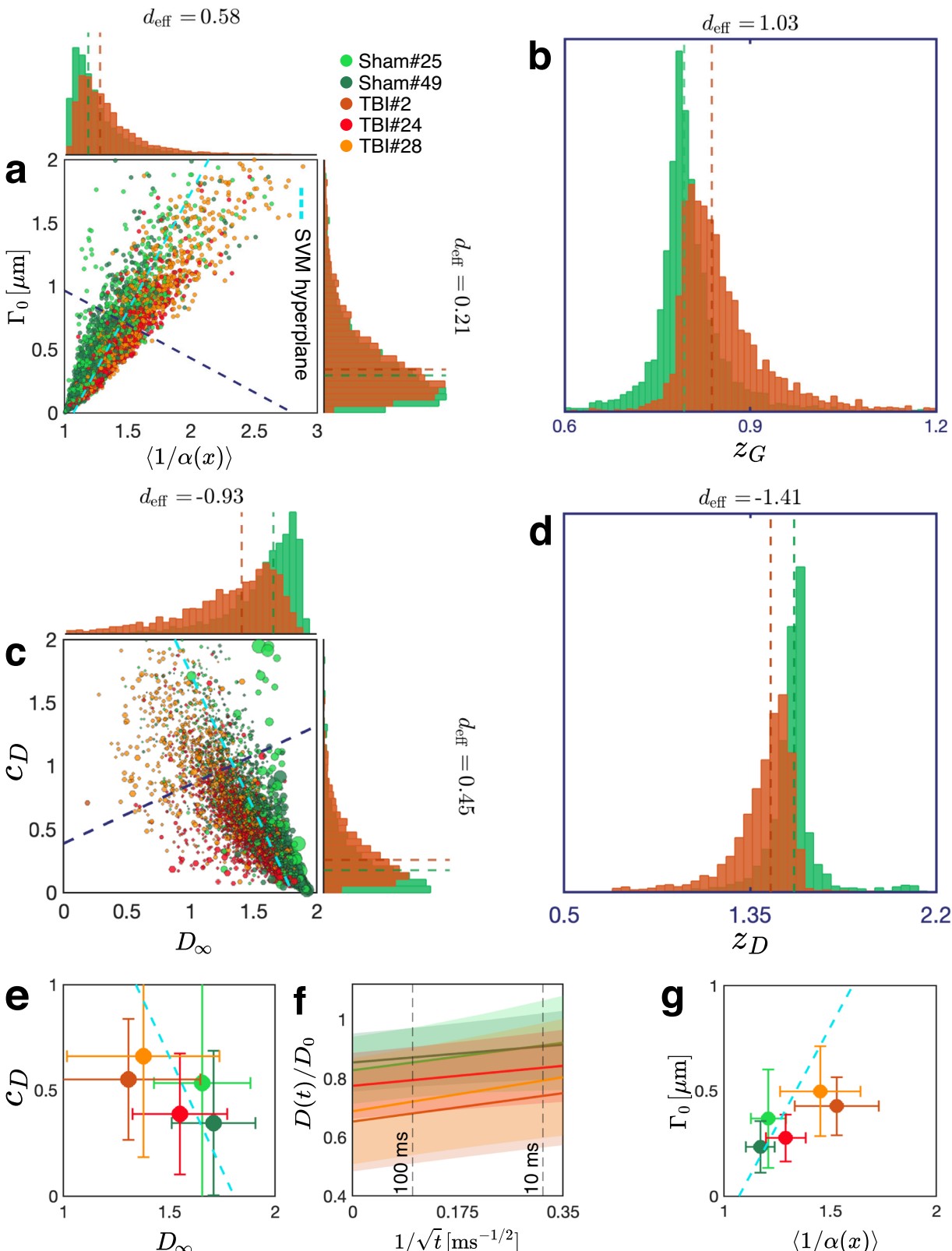

diffusion length $\ell(t)$ with time, as illustrated in Fig. 2d: As time progresses, $t_0 < t_1 < t_2 < t_3 < \infty$, molecules sample larger local microenvironments, homogenizing their statistical properties, such that an axon appears increasingly more uniform. This is equivalent to suppressing Fourier harmonics $\eta(q)$ for $q \gtrsim 1/\ell(t)$. Hence, it is only the $q \to 0$ plateau $\Gamma_0$ of the power spectral density that "survives" for arbitrarily

long $t$ and governs the asymptotic dynamics (1) (provided that the disorder in $\alpha(x)$ is short-ranged, which directly follows from finite $\Gamma_0$, Fig. 2a). The scattering problem is solved in *Methods* in three steps: (i) coarse-graining of the 3-dimensional diffusion equation in a random tube over its cross-section to obtain the one-dimensional (1d) Fick-

**Fig. 3 | Effect of chronic TBI on axon morphology and $D(t)$. a** Geometric tortuosity $\langle 1/\alpha \rangle$, Eq. (2), and the variance $\Gamma_0$ of long-range cross-sectional fluctuations entering Eq. (3), are plotted for myelinated axons segmented from the ipsilateral cingulum of sham-operated (shades of green; $N_{axon}$ = 3999) and TBI (shades of red; $N_{axon}$ = 3999) rats. **b** The optimal linear combination $z_G$ of the morphological parameters is derived from a trained support vector machine (SVM). Projecting the points onto the dark blue dashed line in **(a)** perpendicular to the SVM hyperplane constitutes the maximal separation between the two groups. **c** Predicted individual axon diffusion parameters $D_{\infty,i}$ and $c_{D,i}$ from Eqs. (2)–(3) plotted for myelinated axons in **(a)**. The size of each point reflects its weight $w_i$ in the net dMRI-accessible $D(t)$, proportional to the axon volume. **d** The optimal SVM-based linear combination $z_D$ of the diffusion parameters is derived by projecting the points onto the dark blue dashed line in **(c)** perpendicular to the corresponding SVM hyperplane. Dashed lines in **(a–d)** indicate the medians of the distributions. **e** Macroscopic

diffusivity parameters $c_D$ and $D_\infty$ for each animal are obtained by volume-weighting (filled circles; $N$ = 2 sham-operated and $N$ = 3 TBI) the individual axonal contributions $D_{\infty,i}$ and $c_{D,i}$. Error bars represent measurement uncertainties in the volume-weighted estimates (see *Methods*). The SVM hyperplane (cyan dashed line) is the same as that for the diffusion parameters of individual axons in **(c)**. **f** Predicting the along-tract $D(t)/D_0$ as a function of $1/\sqrt{t}$, Eq. (1), based on the overall $D_\infty$ and $c_D$ in **(e)**. **g** The effect of TBI on the ensemble-averaged geometry (filled circles) is illustrated by transforming the macroscopic ensemble diffusivity in **(e,f)**, as if from an MRI measurement, back onto the space of morphological parameters $\langle 1/\alpha \rangle$ and $\Gamma_0$, via inverting Eqs. (2)–(3). The SVM hyperplane (cyan dashed line) is the same as that for the morphological parameters of individual axons in **(a)**. Error bars corresponding to standard deviations of $D(t)/D_0$ in **(f)** and $\langle 1/\alpha \rangle$ and $\Gamma_0$ in **(g)** are calculated based on errors in **(e)** (see *Methods*). Source data are provided as a Source Data file.

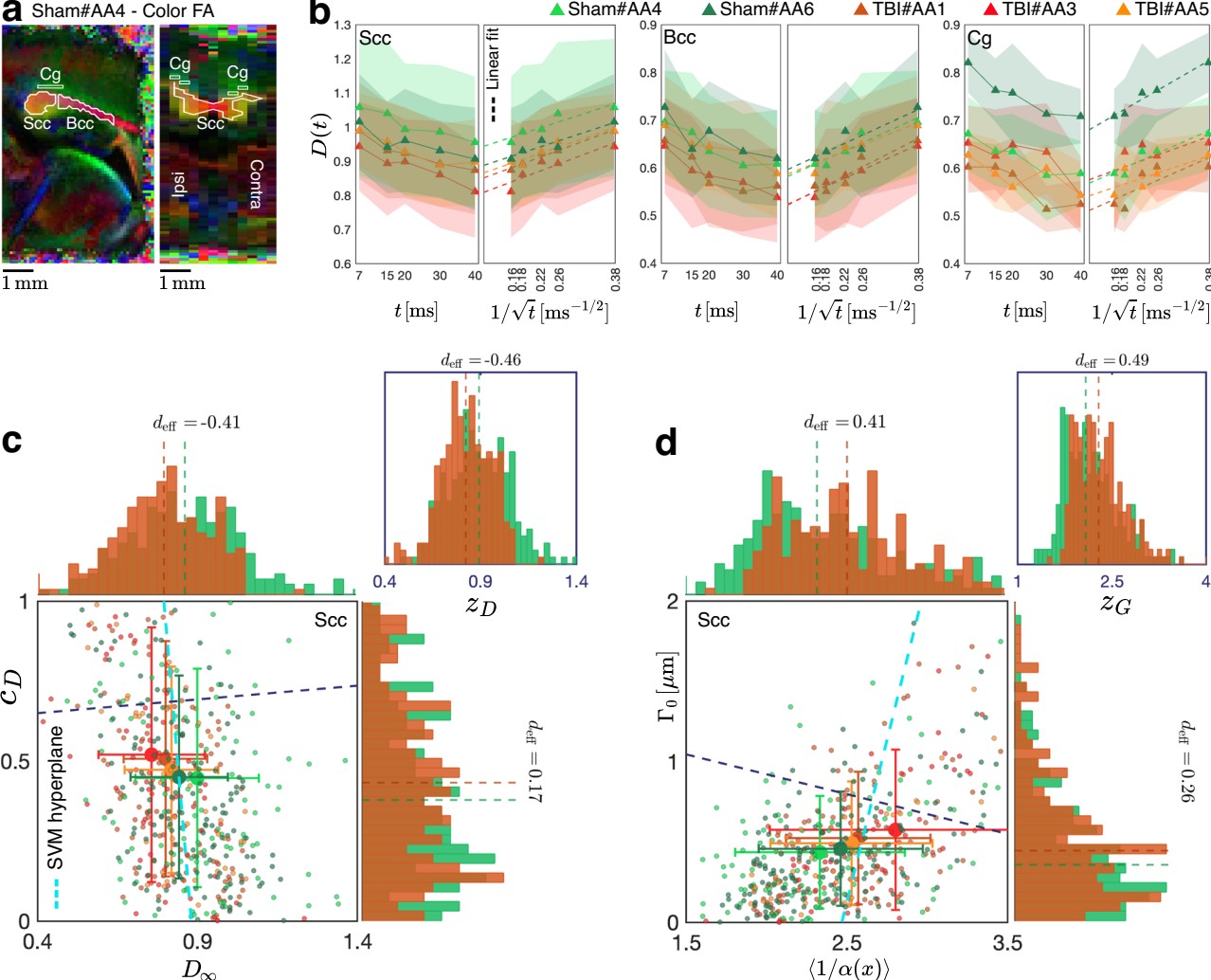

**Fig. 4 | Effect of mild TBI on ex vivo dMRI and axon morphology in ipsilateral major white matter tracts of male rat. a** Representative colored fractional anisotropy (FA) maps in sagittal and coronal views, with the cingulum (Cg), splenium of the corpus callosum (Scc), and body of the corpus callosum (Bcc) annotated. **b** Experimental axial DTI diffusivity $D(t)$ plotted as a function of $t$ and $1/\sqrt{t}$, showing a power-law relation in all ipsilateral white matter regions of interest (ROIs). **c** Diffusion parameters $D_\infty$ and $c_D$ extracted by linear regression of $D(t)$ with respect to $1/\sqrt{t}$ in **(b)** for voxels within the ipsilateral Scc ROI ($N_{voxel}$ = 245 per group). The optimal SVM-based linear combination $z_D$ of the diffusion parameters is derived by projecting the points onto the dark blue dashed line perpendicular to the

corresponding SVM hyperplane. **d** Corresponding geometric parameters $\langle 1/\alpha \rangle$ and $\Gamma_0$, computed by inverting Eqs. (2)–(3) from the diffusion parameters in **(c)**, plotted for voxels in Scc. The optimal linear combination $z_G$ of the morphological parameters is obtained by projecting the data points onto the dark blue dashed line, which is orthogonal to the SVM hyperplane. In **(b)**, filled triangles with shaded areas indicate the mean and standard deviation across the ROI ($N$ = 2 sham-operated and $N$ = 3 TBI). In **(c)-(d)**, each point represents a voxel. Filled circles with error bars indicate the mean and standard deviation across the ROI. Dashed vertical lines overlaid on the distributions denote their medians. Source data are provided in the Source Data file.

Jacobs (FJ) equation[53]

$$\partial_t \psi(t,x) = D_0\, \partial_x \left( A(x)\, \partial_x\, \frac{\psi(t,x)}{A(x)} \right) \qquad (4)$$

with arbitrary stochastic $A(x)$, valid for times $t$ exceeding the time to traverse the cross-section ($t \gtrsim 1\,\mathrm{ms}$); (ii) finding the fundamental solution (Green's function) of Eq. (4) for a particular configuration of $A(x)$; and (iii) disorder-averaging over the distribution of $A(x)$. Step (iii) gives rise to the translation-invariant Green's function $G(\omega, q) = 1/[-i\omega + D_\infty q^2 - \Sigma(\omega, q)]$ in the Fourier domain of frequency $\omega$ and wavevector $q$. Steps (ii) and (iii) are fulfilled by summing Feynman diagrams (Fig. 5) representing individual "scattering events" off the cross-sectional variations $\ln\alpha(x)$, which after coarse-graining over sufficiently long $\ell(t)$ become small to yield the self-energy part $\Sigma(\omega, q)$ asymptotically exact in the limit $\omega, q \to 0$ with $D_\infty q^2/\omega \to 0$. The dispersive diffusivity[37,43,54] $\mathcal{D}(\omega) \simeq D_\infty + \frac{\sqrt{\pi}}{2} c_D \sqrt{-i\omega}$ follows from the pole of $G(\omega, q)$ upon expanding $\Sigma(\omega, q)$ up to $q^2$, yielding Eq. (1) via effective medium theory[52,55]. In *Methods*, we also derive the power law tails $\omega^\vartheta \sim t^{-\vartheta}$ of diffusive metrics for other universality classes of structural fluctuations $\Gamma_\eta(q) \sim |q|^p$, relating the structural exponent $p$ to the FJ dynamical exponent $\vartheta = (p + 1)/2$, with $p = 0$ (short-range disorder) relevant for the axons.

*Undulations*[56] (wave-like variations of axon skeleton) cause a slower, $\sim 1/t$ tail[57] in $D(t)$, a sub-leading correction to Eq. (1). In Eq. (25) of *Methods*, we argue that their net effect is the renormalization of $D(t)$ by $1/\xi^2$, where the sinuosity $\xi \geq 1$ is the ratio of the arc to Euclidean length ($\xi - 1 \sim 0.01 - 0.05$, Supplementary Fig. S11 and Eq. (S23)). All subsequent analysis implies the statistics of cross-sectional variations along the arc-length (see *Methods*), with subsequent rescaling by $1/\xi^2$.

## Validation in axons segmented from volume EM

We now consider the case of chronic TBI (five months post-injury)[16,58,59], both to validate the above theory in a realistic setting and to show how it helps quantify axon morphology changes due to injury. Severe TBI was induced in three rats, and two rats were sham-operated (see *Methods*); animals were sacrificed, and SBEM was performed five months after the sham or TBI procedure. The SBEM datasets were acquired from big tissue volumes of $200 \times 100 \times 65\,\mu\mathrm{m}^3$, with 2/3 of each volume corresponding to the corpus callosum and 1/3 to the cingulum. Samples were collected ipsi- and contralaterally (Supplementary Table S1). We applied the DeepACSON pipeline[16] that combines convolutional neural networks and a tailored tubular decomposition technique[60] to segment the large field-of-view SBEM datasets (Fig. 1). Considering the clinically feasible, long diffusion time asymptote (1), we focus on sufficiently long axons: myelinated axons from the cingulum with a length $L \geq 40\,\mu\mathrm{m}$, and from the corpus callosum with $L \geq 70\,\mu\mathrm{m}$, yielding a total of $N_{\mathrm{axon}} = 36{,}363$ myelinated axons.

To validate the theory, Eqs. (1)–(3), we performed Monte Carlo (MC) simulations using the realistic microstructure simulator (RMS) package[61] in 100 SBEM-segmented axons (43 myelinated axons randomly sampled from two sham-operated rats and 57 from three TBI rats), cf. Fig. 2 and *Methods*. As the sample size limits the lengths of SBEM axons, we also created 50 $L = 500\,\mu\mathrm{m}$-long synthetic axons with statistics of $A(x)$ similar to that in SBEM-segmented axons (see *Methods*) to access smaller Fourier harmonics $q$ for reducing errors in estimating $\Gamma_0$. MC-simulated along-axon diffusion coefficients $D_i(t)$ in the $i$-th axon were volume-weighted (corresponding to spins' contributions to the dMRI signal) to produce the ensemble-averaged $D(t) = \sum w_i D_i(t)$ for each of the synthetic, sham-operated, and TBI populations, where weights $w_i$ are proportional to axon volumes, $w_i \propto \bar{A}_i L_i$, and add up to $\sum_i w_i \equiv 1$, Fig. 2b. The asymptotic form, Eq. (1), becomes evident by replotting $D(t)$ as a function of $1/\sqrt{t}$, Fig. 2c. The individual $D_i(t)$ also exhibits this scaling, albeit with larger MC noise.

Having validated the functional form (1), we used it to estimate and validate $D_{\infty,i}$ and $c_{D,i}$ for individual axons. For that, we employed linear regression with respect to $1/\sqrt{t}$ for $t$ between 10 and 500 ms. Figure 2e validates Eq. (2) for individual axons. Nearly no deviations occur from the identity line for both synthetic and SBEM-segmented axons, indicating the accuracy and robustness in predicting $D_\infty$, given the axonal cross-section $A(x)$.

To validate Eq. (3), we calculated the theoretical value of $c_D$ by estimating the plateau $\Gamma_0$ of the power-spectral density $\Gamma_\eta(q)$ for individual synthetic and SBEM-segmented axons, as shown in Fig. 2a, d, and Supplementary Fig. S1. We then confirmed the agreement between $c_{D,i}$ from MC-simulated $D_i(t)$ for individual axons and their theoretical prediction (3), Fig. 2f, where data points align with the identity line, indicating the absence of bias in the prediction. Random errors in these plots come from errors in estimating $\Gamma_0$, especially for short (SBEM-segmented) axons, as well as from estimating the slope $c_D$ in asymptotic dependence (1) due to MC noise. The numerical agreement with Eq. (3) is notably better for longer synthetic axons. Figure 2f also indicates that as the cross-sectional variation var $[\alpha(x)]$ increases, the deviations from the identity line become more pronounced, which could be attributed to corrections to FJ equation (4), when the "fast" transverse and "slow" longitudinal dynamics are not fully decoupled.

The validated theory opens up the way to massively speed up the predictions of dMRI measurements and their change in pathology based on 3d segmentations: what would normally require over a year of GPU-powered MC simulations in the realistic microstructure of Fig. 1 for tens of thousands of segmented axons is now predicted in mere seconds on a regular desktop computer by calculating the relevant dMRI parameters using Eqs. (2)–(3) based on axon cross-sections $A(x)$.

## Effects of TBI on axon morphology and diffusion

In Fig. 3, we examine the effects of injury in both the morphological coordinates ($\langle 1/\alpha \rangle$ and $\Gamma_0$) and the diffusion coordinates ($D_\infty$ and $c_D$). These equivalent sets of neuronal damage markers are related via Eqs. (2)–(3).

*Morphological parameters* of the individual axons are shown in Fig. 3a for the ipsilateral cingulum (cf. Supplementary Figs. S2–S4 for other regions). TBI causes an increase in both $\langle 1/\alpha \rangle$ and $\Gamma_0$, and manifests itself as a substantial change (1 median absolute deviation, or MAD) of their optimal support vector machine (SVM) combination $z_G$ (subscript $G$ denotes geometry), Fig. 3b. The geometric parameter $z_G$ shows small variations within the sham-operated and TBI groups and a larger difference between the groups.

*Diffusion parameters* of the axons: $D_\infty$ decreases and $c_D$ increases in TBI (Fig. 3c). Their optimal SVM combination $z_D$ (subscript $D$ stands for diffusion) shows an even larger change, 1.4 MAD in TBI, Fig. 3d. Note that the individual $D_\infty$ and $c_D$ are rescaled by the corresponding sinuosity, Supplementary Eq. (S23). As Supplementary Fig. S11 shows, sinuosity increases in TBI.

Due to large sample sizes in panels a and c, we avoid traditional hypothesis testing, which is prone to Type I errors. Hence, we use a nonparametric measure $d_{\mathrm{eff}}$ of the effect size, defined as the difference between the medians normalized by the pooled MAD (Eq. (26) in *Methods*).

The intra-axonal along-tract ensemble diffusivity (1) with volume-weighted $D_\infty = \sum_i w_i D_{\infty,i}$ and $c_D = \sum_i w_i c_{D,i}$, shown in Fig. 3e, f, is predicted based on $D(t)$ originating from five distinct voxel-like sub-volumes in five animals with aligned impermeable myelinated axons. The same SVM hyperplane that separates volume-weighted diffusion parameters of individual axons in Fig. 3c, also separates their ensemble-averaged parameters. Plugging the $c_D$ and $D_\infty$ values into Eq. (1), we predict the associated $D(t)$ for these five voxels as a function of $1/\sqrt{t}$ in Fig. 3f. This representation mimics an axial diffusion tensor

eigenvalue of the intra-axonal space within a coherent fiber tract, demonstrating how MRI can capture TBI-related geometric changes in each voxel.

Finally, we invert Eqs. (2)–(3) to interpret the ensemble-averaged $D(t)$ in terms of the ensemble-averaged morphological coordinates. The separability between the groups is clearly manifest; furthermore, the five points in Fig. 3g, derived from inverting the volume-weighted diffusion parameters in Fig. 3e, are still separable by the same SVM hyperplane that distinguishes the geometrical parameters of individual axons in Fig. 3a.

Based on Fig. 3, we make two observations. (i) The volume-weighting of individual axon contributions in the dMRI-accessible $D(t)$, Fig. 3c, *magnifies* the TBI effect size as compared to the morphological analysis (Fig. 3a). This can be rationalized by noting that TBI preferentially reduces the radii of thicker axons (Supplementary Fig. S8, top row). Hence, the weights $w_i$ change in TBI to emphasize thinner axons, which tend to have greater cross-sectional variations (hence, lower $D_\infty$ and higher $c_D$). (ii) While $\langle 1/\alpha \rangle$ exhibits higher sensitivity than $\Gamma_0$, it is the *two-dimensional parameter space* derived from the time-dependent diffusion (1) that yields the largest effect size for the optimal pathology marker $z_G$ or $z_D$.

### Effect of mild TBI on time-dependent axial $D(t)$ and on axon morphology from ex vivo DTI

We now examine the effects of brain injury in an experimental dMRI setting (Fig. 4 and Supplementary Figs. S5–S7) and compare the results with our predictions based on theory and SBEM segmentations. We measured the ex vivo axial diffusivity $D(t)$ from time-dependent diffusion tensor imaging (DTI) in the white matter of two sham-operated rats and three rats with mild TBI at four weeks post-injury, using a monopolar pulsed gradient spin-echo (PGSE) sequence (cf. *Methods* section). In interpreting these measurements, we assume the dominant contribution to the along-tract diffusion tensor eigenvalue $D(t)$ originates from intra-axonal water within a coherent fiber bundle, in agreement with Eq. (1), as justified in *Discussion* section below.

The axial diffusivity time dependence $D(t)$ is in agreement with the $1/\sqrt{t}$ power-law functional form (1) in all white matter regions of interest (ROIs; Fig. 4a), as shown in Fig. 4b, for which $D(t)$ from sham-operated rats consistently lies above that of TBI animals. Decomposing $D(t)$ into its asymptotic diffusivity $D_\infty$ and the amplitude $c_D$ of its $t^{-1/2}$ power-law approach, Fig. 4c shows that TBI rats exhibit lower $D_\infty$ (substantial negative effect size), and higher $c_D$ (positive effect size), as compared to sham-operated rats.

Translating these diffusion parameters to the morphological parameters, by inverting Eqs. (2)–(3) in Fig. 4d, shows that TBI rats exhibit higher tortuosity (substantial positive effect size), and higher $\Gamma_0$ (positive effect size), as compared to sham-operated rats. The SVM combinations of the experimental diffusional parameters $z_D$ and of the predicted geometrical descriptors $z_G$ both amplify the effect sizes for the respective individual metrics, resulting in a change of 0.46 MAD in TBI.

We observe that the TBI-induced changes in along-axon parameters predicted from axonal geometry (Fig. 3e, f) are *qualitatively similar* to those measured via time-dependent DTI along major tracts (Fig. 4b, c): $D_\infty$ decreases and $c_D$ increases in TBI. Likewise, the changes in dMRI-inferred geometric parameters (Fig. 4d) are similar to those independently obtained from SBEM (Fig. 3a, b): both $\langle 1/\alpha \rangle$ and $\Gamma_0$ increase.

### Geometric interpretation of axonal degeneration

We first consider the tortuosity (2), which is always above 1; its excess $\langle 1/\alpha \rangle - 1 = \mathrm{var}\,\alpha + \mathcal{O}(\langle \delta\alpha^3 \rangle)$ is dominated by the variance $\mathrm{var}\,\alpha$ of relative axon cross-sections. This can be seen by expanding the geometric series $1/(1 + \delta\alpha)$ in the deviation $\delta\alpha = \alpha - 1$, and averaging term-by-term, with $\langle \delta\alpha \rangle \equiv 0$. This means that the more irregular (e.g.,

beaded) the axon shape, the greater the tortuosity and the lower its $D_\infty$ relative to the axoplasmic $D_0$. Indeed, in Fig. 2e, axons with larger cross-sectional variations $\mathrm{var}\,\alpha$ have larger tortuosity.

The meaning of the power spectral density plateau $\Gamma_0$ is a bit more subtle: It quantifies the strength of the structural fluctuations *at large spatial scales* (beyond the disorder correlation length). While the fluctuations at all spatial scales together contribute to lowering the ratio $D_\infty/D_0$, only the large-scale portion $\sim \Gamma_0$ of these fluctuations contributes to the increased amplitude $c_D \propto \Gamma_0$ of the time-dependent part of $D(t)$. Further intuition can be gained from a *single-bead model*. Consider the relative cross-sectional area $\alpha(x) = e^{\eta(x)}$, with $\eta(x) = \eta_0 + \sum_m \eta_1(x - x_m)$, as coming from a set of identical "multiplicative" beads with shape $\eta_1(x)$, placed at random positions $x_m$ on top of the constant $\eta_0$. In Supplementary Eq. (S14), for this model we find $\Gamma_0 = (\sigma_a^2/\bar{a})\phi^2$, where $\bar{a}$ and $\sigma_a$ are the mean and standard deviation of the intervals between the positions of successive beads (assuming uncorrelated intervals), and $\phi = \int dx\, \eta_1(x)/\bar{a}$ is a dimensionless "bead fraction". The factor $\sigma_a^2/\bar{a}$ in $\Gamma_0$ quantifies the disorder in the bead positions, while $\phi^2$ quantifies the prominence of the beads. Hence, $\Gamma_0$ decreases when placing the same beads more regularly and increases for more pronounced beads.

Analyzing the origins of $\Gamma_0$ according to the single-bead model in Supplementary Figs. S9–S10, we found that TBI changes the statistics of bead positions, with (i) a decrease in the mean distance $\bar{a}$ between beads, meaning the number of beads per unit length increases, which is in line with the formation of beads[14,15]; and (ii) a decrease in the standard deviation $\sigma_a$ of the bead intervals, i.e., beads become effectively more ordered (Supplementary Fig. S10). Note that the decrease in $\sigma_a^2$ is stronger than the decrease in $\bar{a}$, such that the overall factor $\sigma_a^2/\bar{a}$ decreases. On the other hand, TBI made beads more pronounced, causing an increase in $\phi^2$.

## Discussion

From the neurobiological perspective, mechanical forces inflict damage on axons during immediate injury and trigger a cascade of detrimental effects such as swelling, disconnection, degeneration, or regeneration over time[15]. In particular, swellings, resulting from interruptions and accumulations in axonal transport, often arrange themselves akin to "beads on a string," defining a pathological phenotype known as axonal varicosities[14,62,63]. These phenomena can persist for months or even years post-injury[59,64].

Our approach shows that axons that survived the immediate impact of injury exhibit morphological alterations that persist into the chronic phase – even in mild TBI, as evidenced by measurements at four weeks post-injury. While not immediately obvious to the naked eye, our proposed morphological and diffusional parameters remain remarkably sensitive to neuronal injury. The laterally induced brain injury leads to increased cross-sectional variance $\mathrm{var}[\alpha(x)]$ in both the cingulum and corpus callosum, with a more pronounced effect observed in the cingulum. This can be attributed to our observation of higher directional homogeneity in axons within the cingulum, rendering a bundle with more uniform statistical properties than the corpus callosum. Furthermore, the response to injury remained localized and confined to the ipsilateral side, closer to the site of injury. The contralateral hemisphere, distant from the immediate impact, exhibits only marginal effects, as expected. Detecting such subtle changes in morphology is crucial as they can contribute to axonal dysfunction, such as altered conduction velocity observed in animal models[65–67], which may be further linked to a diverse array of physical and cognitive outcomes, as well as neurodegenerative conditions, including Alzheimer's disease[68] and epilepsy[69].

While it has been observed that the long-time asymptote $D_\infty$ and the amplitude $c_D$ are qualitatively affected by axonal beadings[12,44,49], their exact relation to tissue microgeometry has remained unknown. Our scattering approach solves this fundamental problem for the intra-

axonal space, drastically reducing the number of degrees of freedom required to specify the randomly-looking geometry of an axon from infinitely many parameters to just two geometric parameters: $\langle 1/\alpha \rangle$ and $\Gamma_0$. This two-parameter space can be subsequently used for comparing and interpreting time-dependent diffusion signals in various injury conditions, different neurodegenerative disorders, as well as in development and aging. The slow $t^{-1/2}$ power-law tail dominates the faster-decaying, $\sim t^{-1}$ contributions due to confined geometries[34], undulations[56,57], or structural disorder in higher spatial dimensions[52], and thus can be used to identify the contribution from effectively one-dimensional structurally-disordered neuronal or glial processes. The $t^{-1/2}$ power law tail in Eq. (1) is similar to that found for one-dimensional short-range disorder in local stochastic diffusion coefficient $D(x)$ of the heterogeneous diffusion equation[52], yet it comes from a different dynamical equation (4).

The direct access to $D(t)$ along structurally disordered axons or dendrites, as well as glial cell processes, can be provided by diffusion-weighted spectroscopy[45,46,70–73] of intracellular metabolites, such as N-acetylaspartate (NAA) for the axons in white matter, and dendrites in the gray matter. Likewise, the effect of cross-sectional variations of glial processes can be quantified by measuring $D(t)$ for other metabolites, such as choline. The trace of the time-dependent diffusion tensor $D_{ij}(t)$ for the corresponding intracellular metabolite at long $t$ would yield the time-dependent diffusion coefficient $D(t)$, Eq. (1), along structurally disordered neuronal or glial cell processes, independent of their orientational dispersion. The time dependence of the metabolite diffusion tensor can be measured using either PGSE or stimulated echo in the time domain[45,46,73], or, equivalently, using oscillating gradients[43,74,75] in the frequency domain.

For water dMRI, our theory is formulated explicitly for intra-axonal diffusion, assuming geometrically disordered, impermeable axons, and does not incorporate signal contributions from extra-axonal space, myelin, or inter-compartmental exchange. One can isolate the intra-axonal $D(t)$ with time-dependent multi-compartment modeling[76], which entails using a set of clinically feasible diffusion weightings and diffusion times, yet requires a high signal-to-noise ratio. Measuring the along-tract diffusion tensor eigenvalue (axial diffusivity) via the lowest-order in diffusion weighting $b \sim q^2$, as in Fig. 4, is much more straightforward, yet it is confounded by the extra-axonal water contribution. This confound would be minimal for highly aligned axonal tracts, with tightly packed axons, where the extra-axonal water fraction is the smallest[77–79], and is further suppressed by the relatively faster extra-axonal $T_2$ relaxation[79–81]. Furthermore, the extra-axonal contribution is qualitatively and quantitatively similar[82], given that its geometric profile mirrors that of intra-axonal space. The myelin compartment contributes minimally due to its low water content and short $T_2$ relaxation time[83]. The relative effect of misalignment of axons within a major tract is small as $\langle \sin^2 \theta \rangle \sim 0.1$ for typical orientation dispersion angles $\theta \sim 20°$[49,78,84]; some of this effect has also been accounted for in the present approach by incorporating undulations, which leads to a mere rescaling of the intra-axonal $D(t)$. Increasing the fiber orientation dispersion would increase these confounding effects. Injury-induced extracellular processes, such as inflammation or glial proliferation, can further modulate the extra-axonal contribution.

Remarkably, our ex vivo dMRI experiment shows that the along-tract DTI measurement already captures the effect of mild TBI, Fig. 4. Moreover, the derived geometric parameters fall within the expected ranges calculated from SBEM-based axon reconstructions. This agreement corroborates the above arguments that the intra-axonal contribution to the overall along-tract DTI eigenvalue dominates in highly aligned tracts, at least in the ex vivo setting. This allows us to conclude that the distinct power law behavior (1) coming from intra-axonal diffusion qualitatively determines the functional form of the along-tract time-dependent DTI eigenvalue. Along-tract $D(t)$ can be

accessed via dMRI measured at a set of diffusion times using pulsed gradients[44,49], or at a set of frequencies with oscillating gradients[47,48] accessing the real part $D_\infty + \sqrt{\frac{\pi}{8}} c_D \sqrt{\omega}$ of the dispersive $\mathcal{D}(\omega)$[37,52,85], or for arbitrary waveform via $\mathcal{D}(\omega)$[37].

Estimating the diffusion coefficient $D(t)$ of the intra-axonal space with time-dependent multi-compartment modeling or spectroscopy corresponds to clinically feasible dMRI weightings, as opposed to very strong diffusion gradients available on only a few custom-made scanners required for mapping axon radii[86]. Moreover, in Supplementary Eq. (S7), we show that the tortuosity $\langle 1/\alpha \rangle$ is sensitive to the lower-order moments of axon radius compared to the effective radius $r_{eff}$ measured at very strong gradients[86] transverse to the tract. Hence, the tortuosity better characterizes the bulk of the radius distribution, while $r_{eff}$ is dominated by its tail, Supplementary Fig. S8 and Eq. (S2). The present approach can be further incorporated into time-varying blocks of multiple diffusion encodings[87–89] or diffusion correlation imaging[90–95], thereby helping resolve contributions from multiple tissue compartments, and turning these techniques into deterministic methods to quantify axonal microgeometry and its changes in pathology, development, and aging.

In summary, the developed scattering approach to diffusion in a randomly-shaped tube exactly relates the macroscopic dMRI measurement to the irregular structure of axons. Thereby, it allows us to factor out the diffusion process and to reveal the structural characteristics that usually remain obscured by diffusion, as they only indirectly affect the dMRI signal. The scattering formalism based on summing Feynman diagrams (Fig. 5) is key to solving the problem, since it allows us to consider the effect of realistic (i.e., structurally random, rather than periodic or fully-restrictive) along-axon geometries.

The exact asymptotic solution of a key dynamical equation (4) radically reduces the dimensionality of the problem: just two geometric parameters not immediately obvious and apparent—average reciprocal cross-sectional area and the variance of long-range cross-sectional fluctuations—embody the specificity of a bulk MRI measurement to changes of axon microstructure in TBI, and enable a near-instantaneous prediction of axial diffusion of the intra-axonal space from tens of thousands of axons. This prediction is further corroborated by an ex vivo dMRI measurement in a male rat model of mild TBI. These two relevant parameters are sensitive to the variation of the cross-sectional area and the statistics of bead positions, opening a non-invasive window into axon shape alterations three orders of magnitude below the MRI resolution.

As an outlook, the present approach combines the unique strengths of machine learning (neural networks for segmentation of large SBEM datasets) and theoretical physics (identifying relevant degrees of freedom) to uncover the information content of diagnostic imaging orders of magnitude below the resolution. This key modeling building block can be used to detect previously established $\mu$m-scale changes that occur not only in axons but also in the morphology of dendrites during aging[96,97], and pathologies such as stroke[98], Alzheimer's[99] and Parkinson's[100,101] diseases, and amyotrophic lateral sclerosis[102,103], with an overarching aim of turning MRI into a non-invasive in vivo tissue microscope.

## Methods
### Animal model and SBEM imaging
We utilized five adult male Sprague-Dawley rats (Harlan Netherlands B.V., Horst, Netherlands; weighing between 320 and 380 g and aged ten weeks). The rats were individually housed in a controlled environment with a 12-hour light/dark cycle and had unrestricted access to food and water. All animal procedures were approved by the Animal Care and Use Committee of the Provincial Government of Southern Finland and performed according to the European Community Council Directive 86/609/EEC guidelines.

TBI was induced in three rats using the lateral fluid percussion injury method described in ref. 104. The rats were anesthetized, and a craniectomy with a 5 mm diameter was performed between bregma and lambda on the left convexity. Lateral fluid percussion injury induced a severe injury at the exposed intact dura. Two rats underwent a sham operation that involved all surgical procedures except the impact. After five months following TBI or sham operation, rats were transcardially perfused, and their brains were extracted and post-fixed. Using a vibrating blade microtome, the brains were sectioned into 1-mm thick coronal sections. From each brain, sections located at −3.80 mm from the bregma were chosen and further dissected into smaller samples containing the regions of interest. Figure 1a shows a sham-operated rat's contralateral hemisphere and the TBI rat's ipsilateral hemisphere. We collected two samples for each brain: the ipsilateral and contralateral samples, including the cingulum and corpus callosum. The samples were stained following an enhanced protocol with heavy metals[105] (Fig. 1b). After sample selection, the blocks were trimmed into pyramidal shapes, ensuring block stability in the microscope sectioning process (For further animal model and tissue preparation details, see ref. 106).

The blocks were imaged using the SBEM technique[107] (Quanta 250 Field Emission Gun; FEI Co., Hillsboro, OR, USA, with 3View). For that, each block was positioned with its face in the $x - y$ plane, and the cutting was done in the $z$ direction. Images were consistently captured with the voxel size of $50 \times 50 \times 50$ nm$^3$ from a large field-of-view $200 \times 100 \times 65$ μm$^3$ at a specific location in the white matter of both sham-operated and TBI animals in both hemispheres. We used Microscopy Image Browser[108] (MIB; http://mib.helsinki.fi) to align the SBEM images. We aligned the images by measuring the translation between the consecutive SBEM images using the cross-correlation cost function (MIB, Drift Correction)[109]. We acquired a series of shift values in $x$ direction and a series of shift values in $y$ direction. The running average of the shift values (window size was 25) was subtracted from each series to preserve the orientation of myelinated axons. We applied contrast normalization such that the mean and standard deviation of the histogram of each image match the mean and standard deviation of the whole image stack. The volume sizes of the acquired EM datasets are provided in Supplementary Table S1.

## Segmentation of myelinated axons

We used the DeepACSON pipeline, a deep neural network-based automaticsegmentation of axons[16] to segment the acquired large field-of-view low-resolution SBEM images. This pipeline addresses the challenges posed by severe membrane discontinuities, which are inescapable with low-resolution imaging of tens of thousands of myelinated axons. It combines the current deep learning-based semantic segmentation methods with a shape decomposition technique[60] to achieve instance segmentation, taking advantage of prior knowledge about axon geometry. The instance segmentation approach in DeepACSON adopts a top-down perspective, i.e., under-segmentation and subsequent split, based on the tubularity of the shape of axons, decomposing under-segmented axons into their individual components.

In our analysis, we only included axons that were longer than $70 \mu m$ in the corpus callosum and 40 μm in the cingulum. We further excluded axons with protrusions causing bifurcation in the axonal skeleton and axons with narrow necks with a cross-sectional area smaller than nine voxels for MC simulations.

## Synthetic axon generation

To generate axons with randomly positioned beads, the varying area $A(x)$ was calculated by convolving the random number density $n(x)$ of restrictions along the line $x$ with a Gaussian kernel of width $\sigma_1$ representing a "bead":

$$A(x) = A_0 + n(x) * A_1 \frac{e^{-x^2/2\sigma_1^2}}{\sqrt{2\pi\sigma_1^2}}, \qquad (5)$$

where we fixed $A_0 = \pi \cdot (0.5)^2$ μm$^2$, let the bead amplitude $A_1$ range between [0.1, 2.5] μm$^2$, and the bead width $\sigma_1$ between [3, 7] μm. The random bead placement $n(x)$ was generated to have a normally distributed inter-bead distance $a$ with a mean $\bar{a} \equiv 1/\langle n(x) \rangle$ ranging between [3, 7] μm and a standard deviation $\sigma_a$ in the range $[0.8 \cdot \bar{a}, 1.2 \cdot \bar{a}]$. The parameters were set to vary in broader ranges compared to refs. 49,50 to cover a broader range of potential axonal geometries.

## Monte Carlo simulations

Monte Carlo simulations of random walkers were performed using the Realistic Monte Carlo Simulations (RMS) package[61] implemented in CUDA C++ for diffusion in a continuous space within the segmented intra-axonal space geometries as described in ref. 49. Random walkers explore the geometry of intra-axonal spaces; when a walker encounters cell membranes, the walker is elastically reflected and does not permeate. The top and bottom faces of each IAS binary mask, artificially made due to the length truncation, were extended with their reflective copies (mirroring boundary condition) to avoid geometrical discontinuity in diffusion simulations. In our simulations, each random walker diffused with a step duration $\delta t = 8.74 \times 10^{-5}$ ms and step length $\sqrt{6D_0 \delta t} = 0.0324 \mu m$ for the maximal diffusion time $t = 500$ ms, with $2 \times 10^5$ walkers per axon. For all our simulations, we set the intrinsic diffusivity $D_0 = 2$ μm$^2$/ms in agreement with the recent in vivo experiments[110].

The time complexity of the simulator, i.e., the number of basic arithmetic operations performed, linearly increases with the diffusion time and the number of random walkers. We ran the simulations on an NVIDIA Tesla V100 GPU at the NYU Langone Health BigPurple high-performance computing cluster. In our settings, the average simulation time within a single intra-axonal space was 16 min, corresponding to about 90 axons per 24 h, such that 36, 363 axons considered in this work would take over 13 months to simulate.

## Fick-Jacobs equation

In what follows, we assume diffusion in a *straight* tube aligned along $x$, with varying cross-section $A(x)$ along its length, and relate the diffusion coefficient $D(t)$ to the statistics of $A(x)$. The case of long-wave undulations on top of the cross-sectional variations will be considered later, in Sec. *Effect of undulations* and in Supplementary Section *The harmonic undulation model*.

Microscopically, the evolution of a three-dimensional particle density $\Psi(t; x, \mathbf{r}_\perp)$ is governed by the diffusion equation

$$\partial_t \Psi = D_0 \nabla^2 \Psi, \quad \nabla^2 = \partial_x^2 + \partial_{\mathbf{r}_\perp}^2, \qquad (6)$$

with a 3d Laplace operator $\nabla^2$, and the boundary condition of zero particle flux through the tube walls. We are interested in integrating out the "fast" transverse degrees of freedom $\mathbf{r}_\perp$ and deriving the "slow" effective 1d dynamics for times $t \gg A(x)/D_0$ over which the density across the transverse dimensions $\mathbf{r}_\perp$ equilibrates. In this regime, $\Psi(t; x, \mathbf{r}_\perp) \simeq \Psi(t, x)$ becomes independent of $\mathbf{r}_\perp$, and the dynamics is described in terms of the 1d density

$$\psi(x) = \Psi(x) A(x). \qquad (7)$$

This implies the adiabaticity of $A(x)$ varying slowly on the scale of a typical axon radius $\sqrt{A/\pi}$. Under these assumptions, the 1d current

density

$$J(x) = -D_0 A(x)\,\partial_x\!\left(\frac{\psi(x)}{A(x)}\right) \qquad (8)$$

defines the FJ equation, Eq. (4) in the main text, for $\psi(x,t)$ via the 1d conservation law

$$\partial_t \psi(t,x) = -\partial_x J(x). \qquad (9)$$

**Tortuosity limit $D_\infty$ at $t \to \infty$, Eq. 2**

Analogously to the problem of resistances in series, let us impose a finite 3D density jump $\Delta\Psi$ across the tube length $L$. Splitting the tube into small segments of lengths $l_i$, $\sum_i l_i = L$, full coarse-graining means that the transient processes die out, such that $\partial_t\Psi \equiv 0$, and current in each cross-section $J(x) = J_i = \text{const}$. According to Eq. (8), this current

$$-J = D_0 A_i \frac{\Delta\Psi_i}{l_i} \equiv D_\infty \bar{A}\,\frac{\Delta\Psi}{L} \qquad (10)$$

defines the coarse-grained effective diffusion constant $D_\infty$, much like dc conductivity. Plugging the net jump

$$\Delta\Psi = \sum_i \Delta\Psi_i = \frac{J}{D_0}\sum_i \frac{l_i}{A_i} \equiv \frac{JL}{D_0}\left\langle\frac{1}{A(x)}\right\rangle$$

into Eq. (10), we obtain Eq. (2) from the main text.

**Asymptotic approach of $D_\infty$, Eq. 3**

Let us separate the constant and the spatially varying terms in the FJ equation, Eq. (4):

$$\partial_t\psi(t,x) = D_0\partial_x^2\psi(t,x) - D_0\partial_x[y(x)\psi(t,x)],$$
$$y(x) = \partial_x \ln\alpha(x). \qquad (11)$$

The last term defines the perturbation, Fig. 5a,

$$\mathcal{V}\psi \equiv -D_0\partial_x[y(x)\psi(t,x)]. \qquad (12)$$

The Green's function (the fundamental solution) of Eq. (11) corresponds to the operator inverse

$$(\mathcal{L}_0 - \mathcal{V})^{-1} = G^{(0)} + G^{(0)}\mathcal{V}G^{(0)} + G^{(0)}\mathcal{V}G^{(0)}\mathcal{V}G^{(0)} + \ldots$$

that has a form of the Born series (Fig. 5b). Physically, this series represents a total probability of propagating from $x_1$ to $x_2$ over time $t$ as a sum of mutually exclusive events of propagating without scattering; scattering off the heterogeneities $\alpha(x)$ $n = 1$ time; $n = 2$ times; and so on. Here $\mathcal{L}_0 = \partial_t - D_0\partial_x^2$ is the free diffusion operator, whose inverse defines the Green's function of the free diffusion equation, diagonal in the Fourier domain:

$$G^{(0)} = \mathcal{L}_0^{-1} \quad \to \quad G^{(0)}_{\omega,q} = \frac{1}{-i\omega + D_0 q^2}. \qquad (13)$$

Disorder-averaging of the Born series turns the products $\sim \mathcal{V}\ldots\mathcal{V}$ into $n$-point correlation functions of $y(x)$, Fig. 5b, and makes the resulting propagator translation-invariant. This warrants working in the Fourier domain, such that the perturbation Eq. (12) corresponds to the vertex operator

$$\mathcal{V}_{k_2,k_1} = -iD_0 k_2\, y_{k_2-k_1} \qquad (14)$$

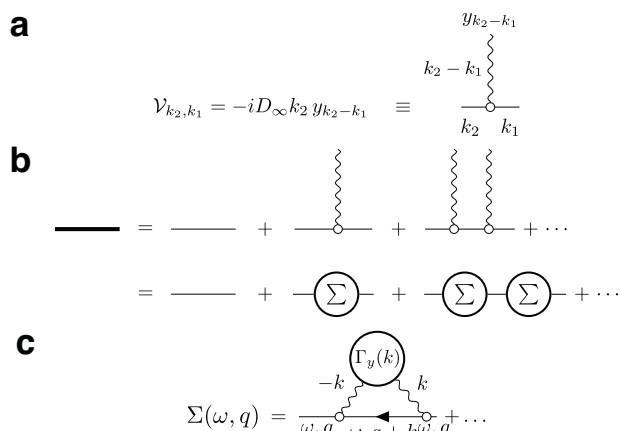

**Fig. 5 | Feynman diagrams for the disorder averaging of the Green's function of Eq. (11). a** The dashed line represents an elementary scattering act off the static disorder potential (12) corresponding to the scattering vertex $\mathcal{V}(\cdot) = -D_\infty\partial_x(y(x)\cdot)$, Eq. (14), where for $t \to \infty$, we substitute $D_0 \to D_\infty$ (see text after Eq. (15)). In the Fourier representation, the scattering momentum (wave vector) is conserved at each scattering event: the sum of incoming momenta ($k_1$ and $k_2 - k_1$) equals the outgoing momentum $k_2$. Since the disorder is static, the "energy" (frequency $\omega$) is conserved in all diagrams. **b** The full Green's function (15), represented by the bold line, is given by the Born series, where propagation between scatterings is described by the free Green's functions $G^{(0)}$, Eq. (13) (thin lines). Averaging over the disorder turns the products $y(x_1)\ldots y(x_n)$ into the corresponding $n$-point correlation functions; the sum of all 1-particle-irreducible diagrams (which cannot be split into two parts by cutting a single $G^{(0)}$ line) is by definition the self-energy part $\Sigma(\omega, q)$. **c** To the lowest (second) order, $\Sigma(\omega, q)$ is given by a single Feynman diagram (16) with the two-point correlation function $\Gamma_y(k)$, Eq. (17).

as shown in Fig. 5a, where the wavy line represents an elementary scattering event with an incoming momentum $k_2 - k_1$ transferred to the particle with momentum $k_1$, such that it proceeds with momentum $k_2$.

According to the effective medium theory formalism (see, e.g., refs. 52,55,111,112), finding the disorder-averaged Green's function

$$G_{\omega,q} = \int dx\, dt\, e^{i\omega t - iqx}G_{t,x} = \frac{1}{-i\omega + D_0 q^2 - \Sigma(\omega, q)} \qquad (15)$$

of Eq. (11) entails summing 1-particle-irreducible Feynman diagrams up to all orders in $\mathcal{V}$, that contribute to the self-energy part $\Sigma(\omega, q)$. This is, in general, impossible analytically.

However, following the intuition of ref. 52, the treatment simplifies in the limit of long $t$ when coarse-graining over a large diffusion length $\ell(t)$ homogenizes the structural disorder $\alpha(x)$, by effectively suppressing its Fourier components $\alpha(q)$ or $y(q)$ with $q \gtrsim 1/\ell(t)$, as schematically depicted in Fig. 2d of the main text. At this point, the original free diffusivity $D_0$ gets renormalized down to $D_\infty$, Eq. (2), and what matters is the *residual* scattering off the long-wavelength heterogeneities; the latter are *suppressed* by the factor $\sim \sqrt{l_c/\ell(t)} \ll 1$ due to coarse-graining over the diffusion length $\ell(t) \gg l_c$ beyond the disorder correlation length $l_c$.

Developing the perturbation theory around the $t \to \infty$ Gaussian fixed point entails changing $D_0 \to D_\infty$ in the free propagator (13) and the scattering vertex (14). Hence, for sufficiently long times $t$, the perturbation (14) can be assumed to be small (essentially, being smoothed over the domains of size $\sim \ell(t)$), and the leading-order correction to the free propagator (13) with $D_0 \to D_\infty$ is determined by the lowest-order contribution to the self-energy part

$$\Sigma(\omega, q) = -D_\infty^2 \int \frac{dk}{2\pi}\,\frac{q(k+q)\,\Gamma_y(k)}{-i\omega + D_\infty(k+q)^2}, \qquad (16)$$

where

$$\Gamma_y(q) = \frac{y(-q)y(q)}{L} \equiv q^2\,\Gamma_\eta(q)\,. \tag{17}$$

Here we introduced

$$\eta(x) = \ln\alpha(x) = \ln\frac{A(x)}{\bar{A}} \tag{18}$$

such that

$$\Gamma_\eta(q) = \frac{\eta(-q)\eta(q)}{L} \tag{19}$$

is its power spectral density. Note that for small variations $\delta\alpha = \alpha - 1$, $\eta(x) = \ln(1 + \delta\alpha(x)) \approx \delta\alpha(x)$ such that $\Gamma_\eta(q) \approx \Gamma_\alpha(q)$ for $q \neq 0$. However, our approach is non-perturbative in $\delta\alpha$ and is valid even for strongly heterogeneous axons.

Finally, we note that the expansion of the term $\Sigma(\omega, q)|_{\omega=0}$ starts with $q^2$ and renormalizes the diffusion constant $D_\infty$ (determined from the dispersion relation $-i\omega + D_\infty q^2 = 0$ defining the low-frequency pole of the propagator (15) up to $q^2$). Hence, in our effective medium treatment, we need to subtract this term from the self-energy part (16). Expanding

$$\Sigma(\omega, q) - \Sigma(\omega, q)|_{\omega=0} \equiv -\delta\mathcal{D}(\omega)q^2 + \mathcal{O}(q^4)$$

provides the dispersive contribution

$$\delta\mathcal{D}(\omega) = -i\omega D_\infty \int \frac{dk}{2\pi}\,\Gamma_\eta(k)\frac{-i\omega + 3D_\infty k^2}{(-i\omega + D_\infty k^2)^2} \tag{20}$$

to the overall low-frequency dispersive diffusivity $\mathcal{D}(\omega) = D_\infty + \delta\mathcal{D}(\omega)$[55]. The corresponding long-time behavior of the *instantaneous* diffusion coefficient[37,55]

$$D_{\text{inst}}(t) \equiv \frac{1}{2}\partial_t\langle x^2(t)\rangle = \int \frac{d\omega}{2\pi}\frac{\mathcal{D}(\omega)}{-i(\omega + i0)}e^{-i\omega t}$$

is found from Eq. (20) by deforming the contour of frequency integration downward from the equator of the Riemann sphere, to pick the 2nd-order residue at $\omega = -iD_\infty k^2$, yielding the long-time tail

$$\frac{\delta D_{\text{inst}}(t)}{D_\infty} = \int \frac{dk}{2\pi}\,\Gamma_\eta(k)\left[1 + 2D_\infty k^2 t\right]e^{-D_\infty k^2 t}$$
$$\equiv \left[1 - 2t\,\partial_t\right]\int \frac{dk}{2\pi}\,\Gamma_\eta(k)\,e^{-D_\infty k^2 t}\,. \tag{21}$$

Note that Eqs. (20)–(21) are valid for any disorder of the tube shape, exemplified by the power spectral density (19). For our case of short-range disorder, defined by the finite plateau $\Gamma_\eta(k)|_{k\to0} = \Gamma_0$, the above equations yield

$$\delta\mathcal{D}(\omega) \simeq \Gamma_0\sqrt{-iD_\infty\omega} \quad \text{and} \quad \delta D_{\text{inst}}(t) \simeq \Gamma_0\sqrt{\frac{D_\infty}{\pi t}}\,.$$

The dispersive $\delta\mathcal{D}(\omega)$ gives the result for $\mathcal{D}(\omega)$ quoted after Eq. (4) in the main text; its real part can be measured with oscillating gradients[37]. The corresponding *cumulative* diffusion coefficient

$$D(t) \equiv \frac{1}{2t}\langle x^2(t)\rangle = \frac{1}{t}\int_0^t dt'\,D_{\text{inst}}(t')$$

measured using pulse-gradient dMRI, acquires the $1/\sqrt{t}$ tail that is double the tail in $D_{\text{inst}}(t)$ above, yielding Eq. (1) with $c_D$ given by Eq. (3).

The above power law tails emerge when $\Gamma_\eta(q) \approx \Gamma_0$ does not appreciably vary on the (small) wavevector scale $0 \leq q \lesssim 1/\ell$, given by the reciprocal of the diffusion length $\ell \sim \sqrt{D_\infty/\omega} \sim \sqrt{D_\infty t}$, i.e., the disorder in $\eta(x)$ has been coarse-grained past its correlation length $l_c$. Equivalently, at such large scales $\ell \gg l_c$, the two-point correlation function

$$\Gamma_\eta(x) \equiv \langle\eta(x)\eta(0)\rangle = \int \frac{dq}{2\pi}e^{iqx}\,\Gamma_\eta(q)$$

of the variations of $\ln\alpha(x)$ can be considered local: $\Gamma_\eta(x) \simeq \Gamma_0\,\delta(x)$.

We also note that the simplification of the self-energy part down to just a single loop, as in Fig. 5c can be formally justified by noting that higher-order contributions bring about higher powers of $\omega$. For example, the lowest-order vertex correction to the self-energy part of Fig. 5c is $\sim i\omega$. This can be seen from the following power-counting argument: each loop brings about the integration that yields an extra small factor

$$\sim \int dk\,(1/k^2)^2 \cdot k^2 \cdot k^2 \sim k \sim \sqrt{i\omega}$$

where (schematically) $1/k^2$ comes from each extra propagator leg, the first factor $k^2$ comes from the two extra vertices (14), and the last factor $k^2$ – from Eq. (17), and we in the end put all momenta on the mass shell $k^2 = i\omega/D_\infty$ since the theory is renormalizable.

## Universality classes of tube shape fluctuations and Fick-Jacobs dynamics

While short-range disorder is most widespread, there exist distinct disorder universality classes[52,113], characterized by the structural exponent $p$ of their power spectral density at low wavevectors $q$, with short-range disorder corresponding to $p = 0$. For our purposes, consider the small-wavevector fluctuations of $\ln\alpha(x)$:

$$\Gamma_\eta(q) \simeq C|q|^p\,, \quad q \to 0\,, \tag{22}$$

where the constant $C$ is the amplitude of the power-law scaling. We can call random tubes with $p > 0$ *hyperuniform*[113] and with $p < 0$ – *hyperfluctuating*. Qualitatively, hyperuniform systems are similar to ordered states with suppressed large-scale fluctuations, whereas hyperfluctuating systems exhibit diverging fluctuations at large scales. Equations (20) and (21) relate the dynamical exponent

$$\vartheta = \frac{p+1}{2} \tag{23}$$

in the power law tails $\omega^\vartheta$ and $t^{-\vartheta}$ of the above diffusive metrics to the structural exponent $p$ in one spatial dimension, generalizing the purely-diffusion theory[52] onto the random FJ dynamics (4). Specifically, for the power spectral density (22),

$$\begin{aligned}
\frac{\delta D_{\text{inst}}(t)}{D_\infty} &= \frac{(1+2\vartheta)\Gamma(\vartheta)}{2\pi}\frac{C}{(D_\infty t)^\vartheta}\,; \\
\frac{\delta\mathcal{D}(\omega)}{D_\infty} &= \frac{(1+2\vartheta)C}{2\sin\pi\vartheta}\left(\frac{-i\omega}{D_\infty}\right)^\vartheta\,; \\
\frac{\delta D(t)}{D_\infty} &= \frac{(1+2\vartheta)\Gamma(\vartheta)}{2\pi(1-\vartheta)}\frac{C}{(D_\infty t)^\vartheta}\,, \quad \vartheta < 1\,,
\end{aligned} \tag{24}$$

where $\Gamma(\vartheta)$ is the Euler's $\Gamma$-function. The last equation, for the $D(t)$ tail, is only valid for sufficiently slow tails, $\vartheta < 1$, corresponding to $p < 1$, i.e., to the tubes where fluctuations are not too suppressed; otherwise, the $1/t$ tail in $D(t)$ will conceive the true $\vartheta$[52]. It is easy to check that for $p = 0$ and $C \to \Gamma_0$, Eqs. (24) correspond to the above results for the short-range disorder.

## Effect of undulations

Let us now consider the effect of long-wavelength undulations (Supplementary Fig. S11) on top of local variations of $A(x)$. Since the undulation wavelength $\lambda \sim 30\,\mu m$[49] is an order of magnitude greater than the correlation length of $A(x)$, in Supplementary Section *The harmonic undulation model* we use this separation of scales to establish that an undulation results in a faster-decaying, $\sim 1/t$ tail in $D(t)$, which is beyond the accuracy of our main result (1) due to the short-range disorder in $A(x)$. Furthermore, the net undulation effect on Eqs. (1)–(3) up to $\mathcal{O}(1/\sqrt{t})$ is in the renormalization of the entire $D(t)$ by the factor $1/\xi^2$. Namely, for the $i$-th axon,

$$D_i(t) = \frac{D_{l,i}(t)}{\xi_i^2} , \qquad (25)$$

Supplementary Eq. (S23), where the sinuosity $\xi_i = L_i/L_{x,i} \geq 1$ is the ratio of the arc to Euclidean length. Here, $D_{l,i}(t)$ is calculated in a "stretched" ("unrolled") axon, i.e., using the arc length $l(x)$ instead of $x$, with $\alpha(x) \rightarrow \alpha(l)$, such that, technically, $\langle 1/\alpha \rangle \equiv \langle 1/\alpha(l) \rangle$, and similar for $\Gamma_0$, in Fig. 2.

Practically, to parameterize the geometry of each axon by its arc length $l$, we evaluate an axonal cross-section $A(l)$ within a plane perpendicular to the axonal skeleton at each point $l(x)$ along its length. This effectively unrolls (stretches) the axon. The values of $A(l)$ along the skeleton are spline-interpolated and then sampled uniformly at $dl = 0.1\,\mu m$ intervals. This equidistant sampling of the curve skeleton ensures a uniformly spaced Fourier conjugate variable $dq$ for calculating the power spectral density $\Gamma_n(q)$.

Axon's volume is determined via $\int dl\, A_i(l) \equiv \bar{A}_i L_i$, which also defines its average cross-sectional area $\bar{A}_i$, where $L_i = \int dl$ is the arc length. This volume measurement is used for volume-weighting the individual axonal contributions $D_i(t)$, Eq. (25), with weights $w_i \propto \bar{A}_i L_i$, to the ensemble diffusion coefficient $D(t)$ along the tract (Supplementary Fig. S12).

## Time-dependent ex vivo dMR imaging

We used five adult male Sprague-Dawley rats (Envigo, Inc., Indianapolis, Indiana, USA), weighing between 338 and 397 g and aged eight weeks at the time of the experiment. The rats were individually housed in a controlled environment with a 12 h light/dark cycle and had unrestricted access to food and water. All animal procedures were approved by the Animal Care and Use Committee of the Provincial Government of Southern Finland and performed according to the European Union Directives 2010/63/EU.

TBI was induced in three rats using the lateral fluid percussion injury method described in ref. 104. Under anesthesia, a 5 mm craniectomy was performed between bregma and lambda on the left convexity, and a mild (1 atm) lateral fluid percussion injury was induced to the exposed intact dura. Two rats underwent sham operations that involved identical surgical procedures except for the fluid impact.

Twenty-eight days after TBI or sham operation, rats were transcardially perfused with 0.9% saline, followed by 2% paraformaldehyde + 2.5% glutaraldehyde in 0.1 M phosphate buffer (pH 7.4) at room temperature for 15 min. Brains were then extracted and post-fixed in 2% paraformaldehyde + 2.5% glutaraldehyde for 2 h at 4 °C, and then placed in 2% paraformaldehyde for further processing.

We performed ex vivo dMRI measurements on a Bruker Avance-III HD 11.7 T spectrometer equipped with a MIC-5 probe giving 3 T/m maximum gradient amplitude on-axis. Before imaging, the brain was coronally sectioned to fit into a 10 mm NMR tube. A 10 mm section of tissue, beginning at the cerebellum contact point and extending anteriorly, was retained, while the cerebellum itself was removed. An additional dorsal cut was made, retaining an 8 mm section, starting from the dorsal surface and extending toward the ventral side. The

brains were then placed in 0.1 M phosphate-buffered saline (PBS, pH 7.4) solution containing 50 $\mu l$/10 ml of gadoteric acid (Dotarem 0.5 M; Guerbet, France) 24 h before scanning to restore $T2$ relaxation properties. MRI was performed on brains immersed in perfluoropolyether (Galden; TMC Industries, United States) inside the NMR tube at 37 °C temperature. Sample temperature was maintained within ±0.25 °C throughout the measurement.

dMRI data were acquired using a 3d segmented PGSE sequence with the following parameters: TR = 1000 ms, TE = 51.26 ms, data matrix $100 \times 100 \times 12$, FOV $9 \times 9 \times 4$ mm, in-plane resolution $90 \times 90\,\mu m^2$, slice thickness 333 $\mu m$ with three segments. A total of 140 diffusion-weighted volumes were acquired, including three sets of 28 uniformly distributed directions at $b$-values of 1000, 2000, and 3000 s/mm², with diffusion gradient parameters $\delta = 2.5$ ms and $\Delta = 7, 15, 20, 30, 40$ ms. Additionally, ten volumes without diffusion weighting ($b = 0$) were collected.

dMRI data were processed following the steps in the DESIGNER pipeline (https://github.com/NYU-DiffusionMRI/DESIGNER-v2)[114] adapted for ex vivo dMRI rat brains. The DESIGNER denoising parameters were set to apply MP-PCA denoising[115,116] with adaptive patch, applying eigenvalue shrinkage[117] and removing partial Fourier-induced Gibbs ringing[118] with 0.69 partial Fourier, followed by Rician bias correction[119]. Diffusion tensor maps were computed using tools provided by the DESIGNER pipeline by fitting the diffusion kurtosis signal representation to account for non-Gaussianity effects.

ROI identification: we assigned slices #2-3 (out of 12 sagittal slices) as contralateral and #10-11 as ipsilateral to the site of injury. We manually drew a bounding box to separate Scc from Bcc and included voxels above 0.5 for Scc and above 0.3 in Bcc in the red channel of the colored FA maps. We applied a lower threshold for Bcc as it is a thinner ROI compared to Scc to include more voxels for statistical hypothesis testing. To draw Cg, we followed the superior boundary of the corpus callosum in the coronal plane and consistently picked a line of voxels above it, guided by the green channel contrast as shown in Fig. 4a.

## Statistics and reproducibility

**Effect size.** We define a nonparametric measure of effect size between two distributions $X$ and $Y$ as

$$d_{\text{eff}} = \frac{Q_{0.5}(X) - Q_0.5(Y)}{\text{PMAD}_{XY}} , \qquad (26)$$

where $Q_{0.5}$ is the median of the distribution, and

$$\text{PMAD}_{XY} = \sqrt{\frac{(n_X - 1)\text{MAD}_X^2 + (n_Y - 1)\text{MAD}_Y^2}{n_X + n_Y - 2}}$$

is the pooled MAD, where MAD calculates the MAD using the Hodges-Lehmann estimator[120].

**Class imbalance.** To address the class imbalance when performing SVMs to separate between sham-operated and TBI axons, we randomly subsampled the dataset with the larger number of axons to match the size of the class with fewer axons.

## Estimation of uncertainties and error bars

**Estimation of error bars.** To quantify uncertainty in diffusion and geometric parameter estimation, we report standard deviations reflecting variability over plausible parameter ranges, rather than statistical fit errors, which capture uncertainty within a fixed model but not variability across modeling choices. In particular:

We performed linear fits to $D(t)$ against $1/\sqrt{t}$ over a range of plausible lower-bound diffusion times $t_0 \in [3, 10]$ ms to determine $D_\infty$ and $c_D$. Horizontal error bars in Fig. 2e, f represent the standard deviations of these fits.

We estimated $\Gamma_0$ by fitting over a range of fractions $\beta \in [0.92\ 0.97]$ of the total variance of the cross-sectional fluctuations (Eq. S1). The resulting standard deviation defines the vertical error bars in Fig. 2f.

In Fig. 3e, error bars represent the weighted standard deviation of $D_{\infty,i}$ and $c_{D,i}$ across individual axons. These errors are then propagated to compute the standard deviations of $\langle 1/\alpha \rangle$ and $\Gamma_0$ shown in Fig. 3g.

**Propagation of uncertainty in linear fits.** To estimate the uncertainty of the predicted line $D(t) = D_{\infty} + c_D/\sqrt{t}$, we used standard error propagation for a linear model: $D(t) = \mu_{D_{\infty}} + \mu_{c_D} \cdot t^{-1/2}$, where $\mu_{c_D}$ and $\mu_{D_{\infty}}$ are the estimated slope and intercept with standard deviations $\sigma_{c_D}$ and $\sigma_{D_{\infty}}$, respectively. The propagated uncertainty of $D(t)$ at a given $t$ is then: $\sigma_{D(t)} = \sqrt{\sigma_{D_{\infty}}^2 + t^{-1} \cdot \sigma_{c_D}^2}$, used for computing the shaded intervals in Fig. 2f and Supplementary Figs. S2–S4.

### Reporting summary

Further information on research design is available in the Nature Portfolio Reporting Summary linked to this article.

## Data availability

Segmentation of white matter microstructure in 3d electron microscopy datasets is publicly available at https://etsin.fairdata.fi/dataset/f8ccc23a-1f1a-4c98-86b7-b63652a809c3[58]. Axonal morphology and time-dependent dMRI data in brain injury that support the findings of this study are publicly available at https://etsin.fairdata.fi/dataset/0e2bc207-0a5a-42f4-b563-1359be9ddf11[121]. Source data are provided with this paper.

## Code availability

The source code of DeepACSON software is publicly available at https://github.com/aAbdz/DeepACSON[122]. The source code of the Monte Carlo simulator (the RMS package) is publicly available at https://github.com/NYU-DiffusionMRI. The source code used to generate the results presented in the manuscript is publicly available at https://github.com/aAbdz/scattering-to-diffusion[123].

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

## Acknowledgements

The research was supported by the NIH under awards R01 NS088040 and R21 NS081230, and by the Irma T. Hirschl fund. This research was supported by the Research Council of Finland (grant #360360) awarded to A.A. The research was performed at the Center of Advanced Imaging Innovation and Research (CAI2R, www.cai2r.net), a Biomedical Technology Resource Center supported by NIBIB with the award P41 EB017183. The dMRI imaging was performed at the Biomedical Imaging Unit of A.I. Virtanen Institute for Molecular Sciences. A.S. was funded by the Research Council of Finland (#323385 and #361370) and Flagship of Advanced Mathematics for Sensing Imaging and Modeling (#358944). H.-H.L was supported by the Office of the Director and NIDCR of NIH with the award DP5 OD031854.

## Author contributions

A.A., E.F., and D.S.N. conceived the project and designed the study. A.A. conducted EM image analysis, performed MC simulations, and analyzed the data. D.S.N. developed theory. R.C.-L. and H.-H.L. contributed to MC simulations. A.S. provided animal models and EM imaging. A.A. provided animal models and dMRI imaging. A.A. and D.S.N. wrote the manuscript. E.F. and D.S.N. supervised the project. All authors commented on and approved the final manuscript.

## Competing interests

The authors declare no competing interests.
