## [Transparent Peer Review file · Nature Communications]

Scattering approach to diffusion quantifies axonal damage in brain injury

Corresponding Author: Dr Ali Abdollahzadeh

Version 1:

Reviewer comments:

Reviewer #1

(Remarks to the Author)
NCOMMS-25-03290A-Z

This is a tour de force study combining innovative scattering physics modeling with post-injury experimental animal axonal data from serial block face scanning electron microscopy reconstructed via convolutional neural networks to determine new metrics of axonal beading and tortuosity that should be measurable using diffusion MRI. The scattering physics model reduces computation time and resources for measuring these axonal properties by many orders of magnitude beyond prior approaches. The spectral results are impressive, showing clear separability of injured axons from sham injury. In chronic traumatic brain injury (TBI), separability is shown to be better in the ipsilateral cingulum than contralateral cingulum, as would be expected. Separability was more challenging in the corpus callosum, however. The ability to detect abnormal axonal beading and tortuosity with diffusion MRI would have wide applicability beyond TBI to a range of neurological diseases, including prevalent neurodegenerative disorders such as Alzheimer disease and Parkinson disease.

The manuscript does have some limitations. Diffusion MRI seems not to have been performed on the experimental animals prior to ultrastructural assessment, therefore actual diffusion MRI measurements of axonal beading and tortuosity that correspond to the theoretical predictions are not provided. The authors also highlight the translational potential of the work; however, no predictions are given for how accurate and reliable these metrics of axonal beading and tortuosity might be when obtained from clinical MRI systems of different field strengths and gradient performance levels using standard diffusion imaging protocols. Finally, the manuscript is written in a style mostly accessible to physicists and I would recommend expanded explanatory sections in which the biological significance and clinical applications are more thoroughly explored, perhaps using terminology and variable names that are more biological and less physical.

(Remarks on code availability)

Reviewer #2

(Remarks to the Author)

The article Scattering approach to diffusion quantifies axonal damage in brain injury develops a new theoretical description of the biophysical relationship between axon morphology features of beaded axons and time-dependent water diffusion along the axon and cross sectional variance. In simulated axons and in those segmented from EM of rat brain tissue after TBI, the theoretical variables are validated. The modeling of beaded axons with a scattering framework using diffusion is creative and interesting and the math seems nicely developed. Additionally, the use of SBEM data from sham and TBI rats was nicely done and offers many advantages over histology-based validation. However, it was challenging to find cohesion between the development of the math, advancement of diffusion MRI biophysical modeling and axonal pathology in TBI. In particular, the main contribution and impact of the work is unclear as written. The text states that the main specific contribution is to link time-dependent diffusion with morphology features of injured axons. If this is indeed the main contribution, there are several concerns listed below that would help to clarify why the development of this biophysical relationship between bead-on-string morphology and $D(t)$ is necessary and about the practicality of using this model on the level of dMRI signal from a voxel. If the main impact is not the biophysical relationships but some aspect of the machine

learning or theoretical approach, the paper should be reframed to make this clear.

Main concerns:

- The relationship between axial diffusivity and bead-on-string axonal damage (which is a well-described outcome of TBI) is well known and well reported in the TBI and other neuroscience literature related to Wallerian degeneration (Examples include Song 2003; MacDonald 2007, but there are many). It has also been extended by characterization of axon diameter distribution (e.g. by DDE) and diffusion correlation imaging which offer improved specificity for beading (For example Komlosh 2018; Benjamini 2020). While the theoretical relationships presented are new and compelling, the idea that diffusion axial to beaded axons is decreased and that cross sectional variance increased is not new. If it is possible to show how the new model can be related to axial diffusivity or some of the other metrics, that might clarify the novelty of the work.
- The analytic biophysical relationship between $D(t)$ and axon morphology as presented seems quite specific to axoplasm morphology only without considering myelin water diffusion or interstitial space. Although interstitial space is mentioned, it is dismissed as the same shape as the axonal space citing Nicholson, although this may not be the case depending on the (de)myelination, edema, gliosis, etc. Side note that EM preparation likely reduces or eliminates interstitial space. Why was the myelin not considered?
- If I understand the description correctly, each axon segmentation or synthetic axon was handled individually, but their relative architecture was not addressed. Since the axons shown are far from parallel to each other and each would have relatively different orientation to any potential diffusion encoding direction, it is hard to understand how the diffusion and morphology parameters formulated for individual axons would be measurable within a voxel where relative axonal geometrics would become averaged. This seems necessary to address.

Minor comments

- Figures and sections are nice
- Writing is good.
- It's a little hard to build an intuition for the variables and follow the math for the main theory (1-3) without a definition for CD which isn't provided in the text. Is it just some variable called this or does it have a biophysical meaning in the model. It is later called one of the diffusion coordinates, but I think for a biophysical model to be well understood it is important to describe each variable, especially each of the 4 meaningful variables in the math.
- Similarly, Γ_0 is a little non-intuitive way when defined (small q plateau of the power spectral density .. cross-sectional fluctuations). Overall it might be helpful to introduce the variables that are most consequential in a bit more detail.

(Remarks on code availability)

Reviewer #3

(Remarks to the Author)

Summary:

This study presents a novel theoretical relationship between two axonal morphology parameters and along-tract diffusion, potentially accessible through diffusion MRI (dMRI). The authors derive this relationship analytically using scattering theory and validate it through simulations based on axonal morphologies extracted from 3D histology. They then apply the framework to diffusion properties observed in rats with traumatic brain injury (TBI), showing that the derived morphological features and corresponding diffusion parameters can differentiate between injured and healthy tissue, highlighting their potential utility as biomarkers.

Strengths:

The key strength of the study lies in its combination of a novel, theoretically derived model and a robust validation demonstrating biomarker potential, within the model's underlying assumptions. The authors present a strong theoretical framework that establishes a principled link between individual axon morphology and the diffusion properties along fiber tracts. The study uses high-resolution 3D histology and deep learning-based segmentation to accurately capture axonal morphology across large ensembles of axons. These realistic axon morphologies serve as the basis for Monte Carlo diffusion simulations, which offer a high-fidelity and biophysically grounded link between morphology and diffusion signal. The progressive validation strategy is fairly convincing: the authors first demonstrate strong correlation between morphological parameters estimated from simulated diffusion MRI and those extracted from histology, then show that axon populations can be discriminated between TBI and sham-operated rats, and finally demonstrate that this separation is preserved after ensemble-level volume averaging, mimicking an MRI-like scenario.

Weaknesses:

Our main concern with the study is whether the proposed model can be translated into practical MRI experiments, given that it relies on histology-based simulations without validation using experimental diffusion MRI data. At the same time, the overall tone (see examples below) occasionally suggests a straightforward translation into practical diffusion MRI measurements without providing an example. Although the authors briefly acknowledge that additional signal contributions are present in water diffusion MRI, this limitation is not fully reflected in the clarity of the wording or the depth of the discussion. For example, the contribution of: the extra-cellular signal, orientation dispersion and crossing fibers to the proposed theoretical signal model are not addressed. For diffusion MR spectroscopy, a realistic discussion of scan time requirements is also missing.

Another concern is the limited dataset. The authors progressively establish validation of the proposed model with Monte Carlo simulations for a small number of axons (100–150). In a second step, they apply the model to larger electron

microscopy (EM) volumes totaling around 36,400 axons across five specimens to demonstrate sensitivity to group differences between TBI and sham-operated rats. While this stepwise approach provides convincing evidence and arguments to justify the progression between stages, the generalizability of the findings to actual MRI voxels remains somewhat limited. The total number of axons included in the study is smaller than what would be found in a single MRI voxel. Hence, it remains questionable whether the presented findings can generalize to the greater structural and pathological heterogeneity present in real-world MRI scenarios, especially considering that the analyzed axons represent a particular subset selected based on length thresholds.

We suggest either (i) a more balanced acknowledgment of the practical challenges in making diffusion MRI measurements specific to the proposed parameters, or (ii) if feasible, some form of validation with experimental diffusion MRI data.

Examples Suggesting Straightforward Translation to MRI:

- Page 4, line 109-111: "The validated theory opens up the way to massively speed up predictions of clinically feasible dMRI measurements and their change in pathology."

- Page 6, line 42-44: "This representation mimics a dMRI measurement, demonstrating how MRI can capture TBI-related geometric changes in each voxel."

- Page 7, line 55-64: "An exact asymptotic solution of a key dynamical equation radically reduces the dimensionality of the problem: just two geometric parameters [...] enable a near-instantaneous prediction of a dMRI measurement from tens of thousands of axons."

- page 7, lines 20 -24: "Accessing the along-tract diffusion coefficient $D(t)$ corresponds to a clinically feasible, lowest-order $\sim q^2$ dMRI weighting, as opposed to very strong diffusion gradients (available on only a few custom-made scanners) required for mapping axon radii." -> But, in your simulations, you do not incorporate: a MRI-representative ensemble of axons; you do not incorporate extra-axonal signal; crossing fibres; fibres with different size (cross-sectional und longitudinal).

Specific questions/suggestions/comments:

- page 4, lines 68-72: "[...]MC-simulated along-axon diffusion coefficients $D_i(t)$ in the i -th axon were volume-weighted (corresponding to spins' contributions to the dMRI signal) to produce the ensemble-averaged $D(t) = \sum w_i D_i(t)$ [...]" Could you introduce the assumptions regarding fiber configuration and signal compartments? In particular, Is it also valid if extracellular signal is present and for crossing fibres (if so please explain why)? Where was this relation derived? Does it come from Lee et al., 2020 (Eq. 10a)?
- Are the sketched axons in Fig. 2a true to scale? And why does the diameter of the synthetic axon appear larger than the one of the segmented axons? Why did you choose $\alpha = -1$ on the y-axis?
- Fig. 2d: how does the Gaussian kernel length in the caption relate to the sketched axons? It might also be helpful to highlight that the right-most color-bar is not another axon.
- Synthetic axons in Fig. 2: Why were those numbers ($L=500 \mu$ and $N=50$) chosen? How do the results change as a function of L ?
- The color coding in Fig. 2e-f may benefit from improved contrast to support clearer differentiation.
- The definition of error bars in Figure 2, 3 and S2-4 could be clarified. Are these errors standard errors arising from linear fits of diffusion parameters and polynomial fit of Γ_0 ? If so, errors could be shown in $D(t)/D(\infty)$ over $1/\sqrt{t}$ graphs as well.
- How do the synthetic axons help "reducing errors in estimating Γ_0 " for EM-based axons? Was the information from synthetic axons directly used for fitting Γ_0 of EM-based axons? See page 4, lines 67-72.
- The authors suggest an absence of bias in Figure 2f. However, there appears to be a slight systematic deviation for synthetic axons, which could also be present but obscured by the high density of data points for EM-segmented axons. Could you provide a plot in which points have less overlap? Could you comment on what might underlie the slight bias?
- Page 4, line 109: "The validated theory opens up the way to massively speed up predictions of clinically feasible dMRI measurements and their change in pathology: [...]" As I understand it, the validated theory does not speed up clinically feasible dMRI measurements, but might potentially enable them in the first place.
- Fig. 3f: Is the observed difference in $D(t)$ statistically relevant? How do the variations in "e" propagate into "f"?
- Page 7, line 35-39: "For water dMRI, we expect the along-axon extra-axonal space contribution to be similar given that its geometric profile mirrors that of intra-axonal space [...]" The statement may overstate the similarity between intra- and extra-axonal contributions. Could you elaborate on why this is your expectation?
- Page 7, line 30-34: The authors introduce the paragraph with the clinical feasibility in terms of "lowest-order q^2 weighting". How do the practical methods mentioned in this paragraph fit into other aspects of clinical feasibility, e.g., scan times, voxel size in diffusion MR spectroscopy?
- Page 6, line 57-59: The authors attribute the stronger effect size in volume-weighted diffusion metrics to axon radius differences. Possibly, imperfect segmentation could confound this observation. Showing not only distributions of axon radii but also including axon length and volume could support the authors' attribution of volume-change mainly to axons radii decrease more unambiguously.
- Page 7 line 3-5: "The repercussions of the laterally induced brain injury affected both the cingulum and corpus callosum, with a pronounced effect on the cingulum." It's unclear to what repercussion in the corpus callosum the authors refer.
- Discussion: the authors don't discuss the impact of axon selection by length on the observed results
- What about inter-individual differences as a confounder? E.g., Fig. S3a-c suggest great similarities between Sham#25 and TBI#28 than within groups.
- Page 6, line 36: "[...] five distinct voxels in five animals with aligned impermeable myelinated axons." What is meant with voxels? How do they relate to MR voxels?
- Page 6, lines 40 - 42: Plugging the cD and D_∞ values into Eq. (1), we predict the associated $D(t)$ for these five voxels as a function of $1/\sqrt{t}$ in Fig. 3d. -> Shouldn't this be Fig. 3f?

- Figure S1: The fitting looks very unstable. Can you motivate why a second order polynomial can be used for the fitting and how the constraints are motivated? How does a potential error in Γ_0 propagate into the group difference analysis in Fig. 3 etc.
- Page 6, line 25: The link to Eq. (S22) is set wrongly.
- Figure S5 caption: "The histograms demonstrate the decrease in the axon radius caused by TBI on the ipsilateral side." This is only true for the cingulum.
- The effect size is defined in terms of the median absolute deviation, but the results section refers to it as "standard deviations" See, e.g., page 6, line 15; and page 6, line 23.
- What is the motivation for the relative sample sizes (57 TBI vs. 43 sham axons)? See page 4, lines 62–67.
- Page 7, lines 23 - 24: "Moreover, in Supplementary Eq. (S7), we show that [...]" -> It is not linked to Eq. S7.
- Equation (22): Please provide intuitive information about "C"

(Remarks on code availability)

Code Availability:

The authors share the code for axon segmentation and Monte Carlo simulations, providing access to the main building blocks of their analysis pipeline. However, the code for reproducing the specific results presented in the manuscript is not publicly available. We have not attempted to test the provided code.

Reviewer #4

(Remarks to the Author)

(Remarks on code availability)

Version 2:

Reviewer comments:

Reviewer #1

(Remarks to the Author)

The authors have satisfied my comments with their revisions.

(Remarks on code availability)

Reviewer #2

(Remarks to the Author)

The revised manuscript Scattering approach to diffusion quantifies axonal damage in brain injury describes an analytic biophysical relationship between axon cross sectional variability and water diffusion providing a bridge between this specific cellular feature and the physical process of water diffusion that is measurable by MRI. The model could be a foundation for more specific non-invasive mapping of axonal features in disease states (e.g. TBI) in the future. The model is validated using Monte Carlo simulations with SBEM data and the additional *ex vivo* diffusion MRI experiments were performed with rat brain tissue after mild TBI to demonstrate how the model may provide axon specific metrics in disease states. The main contribution of the work is the theoretical development of a model for axonal geometry and the simulation and experimental work supporting the model is appropriate. The revised writing is much improved, especially the goals of the work are more clear and the variables of the model are defined in a way that builds an intuitive understanding.

The impact seems modest. While it is not clear that developing this model will have practical implications that improve what already exists for detection of axonal pathology after TBI or in other disease states, the development of the theory is creative and nicely done and the direct analytic bridging of axonal microgeometry and diffusion is compelling. The methodology is sound.

The manuscript and figures are appropriate and I suggest no revision.

(Remarks on code availability)

Reviewer #3

(Remarks to the Author)

We appreciate that the authors have addressed most of our comments and the additional *ex vivo* experiment complemented

and increased the relevance of the paper. In total, there is significant innovation and evidence for the important claims in the paper, and therefore we recommend publication. We would also like to agree with the initial statement of the authors where they highlighted the strength of their paper and emphasize that questioning those contributions are and were not subject of our comments.

However, we would recommend that the limitations of the methods are clearly stated in the discussion and that some of the current statements are toned down accordingly. Please consider the following comments:

1. It seems that the authors agree that it is unclear for in vivo measurements whether signal will be mainly coming from the intra-cellular compartment and thus the estimated axial diffusivity can be interpreted in terms of the proposed modelled signal. This is particularly clear from the answer to R3-2. However, this is not adequately reflected in the discussion. We suggest that parts of the answer go into the discussion, so that not only the reviewers but all readers can benefit from this clarification. Here are the relevant passages:

“Our theoretical model is formulated explicitly for intra-axonal diffusion, assuming geometrically disordered, impermeable axons, and does not incorporate signal contributions from extra-axonal space, myelin, or inter-compartmental exchange. [...] in in vivo dMRI, the measured signal will include contributions from extra-axonal water, and this will numerically affect both cD and D_{∞} . [...]”

In this context, the following statements in the paper need clarification:

- Page 8, lines 60-64: “While it has been observed that the long-time asymptote D_{∞} and the amplitude cD are qualitatively affected by axonal beading [12, 46, 51], their exact relations with tissue microgeometry have remained unknown. Our scattering approach solves this fundamental problem, [...]”

Assuming that there is extra-cellular signal contribution in the estimated diffusion parameters in vivo application, it is questionable that the employed model can explain the “exact relations with tissue microgeometry”.

- Page 8; lines 103-106: “Remarkably, our dMRI experiment shows that even the lowest-order in diffusion weighting $b \sim q^2$, the along-tract DTI, already captures the effect of mild TBI, Fig. 4.” For ex-vivo MRI only.

- Page 9, lines 59 – 63: “These two relevant parameters are sensitive to the variation of the cross-sectional area and the statistics of bead positions, opening a non-invasive window into axon shape alterations three orders of magnitude below the resolution of clinical MRI.”

Clinical MRI is typically applied in vivo, but in vivo the validity of the employed model is unclear due to the unknown extra-cellular signal contribution.

- Page 9, lines 69 – 76: “It can be used to detect previously established μm -scale changes that occur not only in axons but also in the morphology of dendrites during aging [97, 98], and pathologies such as stroke [99], Alzheimer’s [100] and Parkinson’s [101, 102] diseases, and amyotrophic lateral sclerosis [103, 104], with an overarching aim of turning MRI into a non-invasive in vivo tissue microscope.”

Again, the validity of the established relationship for in vivo application is unknown due to the unknown extra-cellular signal contribution.

2. In the discussion of the ex vivo data, there is an interpretation on page 8 (lines 108 – 114) which could be further toned down: “We believe this is because the intra-axonal contribution to the overall along-tract DTI eigenvalue dominates in the highly aligned tracts, while the extra-axonal contribution is also qualitatively and quantitatively similar [78], given that its geometric profile mirrors that of intra-axonal space.”

Also other effects in the extra-cellular signal unrelated to the “geometric profile mirroring that of intra-axonal space” could also lead to a reduction of the diffusion coefficient such as inflammatory processes and thereby potentially also produce a similar behaviour as that one observed experimentally.

3. The mentioning of diffusion-weighted spectroscopy as a useful method for this application, is still unclear to us. For example on page 8, lines 83-87, the authors state: “The most direct and specific access to the time dependent diffusion within the intra-cellular space can be provided by diffusion-weighted spectroscopy [47, 48, 71–74] of intracellular metabolites, such as N-acetylaspartate (NAA) for the axons. The same NAA-based approach can be used to probe the cross-sectional variations of the dendritic processes in the gray matter. [...]”

If we understand the authors correctly, they claim that their model can be applied to the time dependent diffusion data acquired with those NAA-based approaches. But, later in the discussion, the authors highlight that their model is only valid for highly aligned fibres (page 9, lines 3-9). Clearly, in the structures investigated (in vivo) by the papers cited above, the fibres were not highly aligned and in gray matter this assumption is also not valid. Could you please clarify this contradiction?

4. In the discussion the authors sound very optimistic about the in vivo applicability of along-tract $D(t)$ (Page 9, lines 18-22) for the intra-axonal signal and put it above high- b -value measurements in terms of its practicability: “Estimating the diffusion coefficient $D(t)$ of the intra axonal space corresponds to clinically feasible dMRI weightings, as opposed to very strong diffusion gradients (available on only a few custom-made scanners) required for mapping axon radii [87].”

The time-dependent approach is of course of higher practicability than the high- b -value approaches, but as long as the role of the extra-cellular signal contribution for in vivo measurements is not clarified, it is unclear whether the proposed model is superior to the ref approach.

(Remarks on code availability)

Reviewer #4

(Remarks to the Author)

(Remarks on code availability)

Scattering approach to diffusion quantifies axonal damage in brain injury

Ali Abdollahzadeh,^{1,2,*} Ricardo Coronado-Leija,¹ Hong-Hsi Lee,³

Alejandra Sierra,² Els Fieremans,¹ and Dmitry S. Novikov^{1,*}

¹*Center for Biomedical Imaging, Department of Radiology,
New York University School of Medicine, New York, NY, USA*

²*A.I. Virtanen Institute for Molecular Sciences, University of Eastern Finland, Kuopio, Finland*

³*Athinoula A. Martinos Center for Biomedical Imaging, Department of Radiology,
Massachusetts General Hospital, Harvard Medical School, Boston, MA, USA*

Common response to all reviewers

We thank all the reviewers for the time and effort they spent on our manuscript and for their encouraging remarks. From the reviewers' comments, we identified a common theme regarding the lack of experimental time-dependent dMRI demonstration.

We would like to emphasize that this study, at its core, achieves the ultimate goal of dMRI modeling (as applied to the intra-axonal space): *it relates the measurement directly to the tissue structure*, letting us bypass the diffusion-weighted signal. Any such relation is unique, special and represents considerable progress, since we do not observe the cellular microstructure directly in MRI, but rather its fairly indirect and subtle effects on the measured signal.

Technically, we achieved this goal by developing the *analytical framework* for exactly relating the along-axon time-dependent diffusion to the structurally random (irregular) axonal shapes, and for quantifying the relevant parameters of such random shapes in realistic axonal geometries. Note that both characterizing random shapes and modeling diffusion in them are intrinsically more challenging than dealing with regular (periodic or fully restricted) geometries. We furthermore validated the theory through MC simulations using realistic geometries obtained from high-resolution 3d-EM volumes.

The theoretical formalism, involving rewriting the Fick-Jacobs equation in terms of a scattering problem and summing the relevant Feynman diagrams to obtain the self-energy part of the disorder-averaged propagator, is a nontrivial development that enabled solving this previously unsolved yet pivotal problem. The theory offers a closed-form forward model that predicts time-dependent axial diffusivity in the intra-axonal space given the microgeometry of axons and identifies the relevant (and non-obvious) geometric characteristics of the random axonal shapes.

In this regard, discussing other neurodegenerative changes (e.g., demyelination) is conceptually beyond the scope of this work. We are strong believers in building our fundamental understanding step by step, phenomenon by phenomenon, each time standing on firm footing. Remarkably, the intra-axonal geometric properties and the diffusion in the intra-axonal space already distinguish TBI from sham.

We note that the 3d-EM histology used in this study was conducted prior to the advances in time-dependent dMRI techniques developed in this work. Repeating the full experimental pipeline, including the injury model, EM preparation, 3d-EM imaging, and segmentation, would require a substantial investment of time, resources, and funding. Undoubtedly, such an effort would be worthwhile in its own right, but it falls beyond the scope of the present study.

Appreciating the thoughtful reviewer feedback, over the past few months, we conducted a whole **new *ex vivo* time-dependent dMRI experiment (new Fig. 4)** on two sham-operated rats and three rats with mild TBI, sacrificed four weeks post-surgery. Due to the mentioned practical limitations, we were unable to replicate the exact severe TBI model at five months post-injury from the original study. Instead, we opted for a mild TBI model at one month post-injury.

While the injury severity and chronicity differ, we maintained biological consistency by using the same rat strain, adult Sprague-Dawley rats, and induced injury at the same anatomical location. It is conceivable that five months of recovery following a severe TBI may yield a microstructural state comparable to that of mild TBI at one month, owing to tissue remodeling and healing—we fully acknowledge the very complex and multifaceted nature of post-injury brain processes. Though not identical, the animal model for dMRI measurements retains key biological features that allow us to assess the sensitivity of the diffusion parameters to axonal alterations.

These measurements demonstrate that the changes in time-dependent diffusion parameters, c_D (amplitude of the time-dependent part) and D_∞ (long-time limit of the diffusion coefficient), predicted based on the changes in geometrical descriptors Γ_0 and $\langle 1/\alpha(x) \rangle$ from histology in Fig.3, do indeed exhibit the anticipated changes (new Fig.4).

* Corresponding authors:

ali.abdollahzadeh@uef.fi, dmitry.novikov@nyulangone.org

This preliminary experimental evidence, together with the MC simulations and quantification of over 36,000 axons, further corroborates the plausibility and predictive power of our analytical framework.

We added the following sections to the Main text, new Fig. 4 (page 6), Discussion (page 8-9), and Methods (page 18-19) sections. We also **rewrote the Discussion** section; highlighting all the changes there would have resulted in coloring almost all text, hence we highlighted just a few key changes.

We hope the reviewers can appreciate the effort we took to make the presentation that much more comprehensive, as well as to address their other concerns, and that the manuscript could be published in its current form.

REVIEWER COMMENTS

NCOMMS-25-03290A-Z

Response to Reviewer #1

This is a tour de force study combining innovative scattering physics modeling with post-injury experimental animal axonal data from serial block face scanning electron microscopy reconstructed via convolutional neural networks to determine new metrics of axonal beading and tortuosity that should be measurable using diffusion MRI. The scattering physics model reduces computation time and resources for measuring these axonal properties by many orders of magnitude beyond prior approaches. The spectral results are impressive, showing clear separability of injured axons from sham injury. In chronic traumatic brain injury (TBI), separability is shown to be better in the ipsilateral cingulum than contralateral cingulum, as would be expected. Separability was more challenging in the corpus callosum, however. The ability to detect abnormal axonal beading and tortuosity with diffusion MRI would have wide applicability beyond TBI to a range of neurological diseases, including prevalent neurodegenerative disorders such as Alzheimer disease and Parkinson disease.

We thank the reviewer for their positive remarks.

R1-1: The manuscript does have some limitations. Diffusion MRI seems not to have been performed on the experimental animals prior to ultrastructural assessment, therefore actual diffusion MRI measurements of axonal beading and tortuosity that correspond to the theoretical predictions are not provided.”

We have now added a full experimental dMRI section; please refer to “Common response to all reviewers”.

R1-2: The authors also highlight the translational potential of the work; however, no predictions are given for how accurate and reliable these metrics of axonal beading and tortuosity might be when obtained from clinical MRI systems of different field strengths and gradient performance levels using standard diffusion imaging protocols.

The added experimental section shows that the predicted geometric parameters, Γ_0 and $\langle 1/\alpha(x) \rangle$, can indeed be detected already via c_D and D_∞ estimated from time-dependent DTI. The close agreement between histology-predicted (Fig. 3e-f) and measured (Fig. 4b-c) diffusion parameters provides strong support for the practical measurability of these parameters, despite some bias from fiber orientation dispersion and extra-axonal water.

In human subjects, Fieremans *et al* [1] measured the axial diffusivity $D(t)$ in major white matter tracts *in vivo* using STimulated Echo Acquisition Mode (STEAM)-DTI sequences for relatively long diffusion times, $t = 45 - 600$ ms. The study found pronounced time-dependence in the axial diffusivity in both anatomically based white matter regions and FA thresholded regions, observing a consistent power-law decay $t^{-1/2}$, the key signature of the derived time-dependence $D_a(t)$. Further, Lee *et al* [2] demonstrated the same power-law behavior of the time-dependent axial diffusivity $D(t)$ in the range $t = 22 - 100$ ms measured using monopolar pulsed gradient spin echo (PGSE). Both measurements were performed on regular 3T clinical scanners (Siemens Trio and Prisma, respectively). More recently, Arbabi *et al* [3] and Tan *et al* [4] measured the same power law $\sim \omega^{1/2}$ using oscillating gradients in human subjects.

Additionally, metabolite diffusion, especially N-acetylaspartate (NAA; an intracellular neuronal/axonal metabolite), exhibits time-dependent diffusion [5]. While the specific power law in white matter has not yet been published, our research group at NYU Langone Health is actively pursuing this line of work.

These considerations are now added more comprehensively to the Discussion, page 8, lines 83-115, page 9, lines 1-13.

R1-3: Finally, the manuscript is written in a style mostly accessible to physicists and I would recommend expanded explanatory sections in which the biological significance and clinical applications are more thoroughly explored, perhaps using terminology and variable names that are more biological and less physical.

In the revised manuscript, we have expanded the discussion to more explicitly articulate the biological significance and clinical relevance of our findings (Main text, page 7, lines 98-110; Discussion, page 8, lines 63-73). However, the relation of dMRI to microstructure is an essentially condensed matter physics problem, and there is no escape from the physicist’s mindset. We hope our work can serve as a bridge between the physics and neuroimaging communities.

Response to Reviewer #2:

The article Scattering approach to diffusion quantifies axonal damage in brain injury develops a new theoretical description of the biophysical relationship between axon morphology features of beaded axons and time-dependent water diffusion along the axon and cross sectional variance. In simulated axons and in those segmented from EM of rat brain tissue after TBI, the theoretical variables are validated. The modeling of beaded axons with a scattering framework using diffusion is creative and interesting and the math seems nicely developed. Additionally, the use of SBEM data from sham and TBI rats was nicely done and offers many advantaged over histology-based validation.

We thank the reviewer for their positive remarks.

R2-1: However, it was challenging to find cohesion between the development of the math, advancement of diffusion MRI biophysical modeling and axonal pathology in TBI. In particular, the main contribution and impact of the work is unclear as written. The text states that the main specific contribution is to link time-dependent diffusion with morphology features of injured axons. If this is indeed the main contribution, there are several concerns listed below that would help to clarify why the development of this biophysical relationship between bead-on-string morphology and $D(t)$ is necessary and about the practicality of using this model on the level of dMRI signal from a voxel. If the main impact is not the biophysical relationships but some aspect of the machine learning or theoretical approach, the paper should be reframed to make this clear.

The core technical contribution of our work lies in the derivation of an asymptotically exact expression for the time-dependent axial diffusion coefficient $D(t)$ in realistic 3d axon geometries. To achieve this, we develop scattering theory for the Fick-Jacobs equation (diffusion in a tube of varying shape), which involves summing Feynman diagrams to evaluate the self-energy part of the propagator. This extends modern techniques from field theory and condensed matter physics onto the biomedical realm, with an immense practical potential.

This formalism yields a powerful reduction of the complex, infinite-dimensional irregular axonal geometry into merely two diffusion-relevant parameters, which quantify cross-sectional fluctuations of axons. These two parameters uniquely govern $D(t)$ and therefore provide interpretable and measurable links between axonal microgeometry and dMRI observables.

The experimental settings complement the theoretical framework to show that the derived geometrical descriptors—and their diffusion-relevant counterparts—can pick up the differences between sham-operated and TBI rats. In particular, the revised manuscript now includes time-dependent *ex vivo* dMRI measurements (please refer to “Common response to all reviewers”) that independently confirm the same narrative predicted by the theory.

This has been stated in the Abstract on page 1:

“Scattering theory uncovers the two parameters that determine the diffusive dynamics of water long axons: the average reciprocal cross-section and the variance of long-range cross-sectional fluctuations.”

In the Main text on page 1:

“Here, we identify the morphological parameters associated with pathological changes in axons that can be probed with dMRI measurements — thereby establishing the link between cellular-level pathology and noninvasive imaging. Specifically, we analytically connect the axonal microgeometry (Fig. 1) to the time-dependent along-tract diffusion coefficient (Fig. 2)

$$D(t) \equiv \frac{\ell^2(t)}{2t} \simeq D_\infty + \frac{c_D}{\sqrt{t}}, \quad t \gg t_c. \quad (1)$$

As discussed below, $D(t)$ is accessible with dMRI [45-50] as the along-tract diffusion coefficient in the clinically feasible regime of diffusion time t exceeding the correlation time $t_c \sim 1$ ms to diffuse past μm -scale axon heterogeneities.”

In the Conclusions and outlook on page 9, we have now reiterated our contribution more clearly:

“The developed scattering approach to diffusion in a randomly-shaped tube exactly relates the macroscopic diffusion MRI measurement to the irregular structure of axons. Thereby, it allows us to factor out the diffusion process and to reveal the structural characteristics that usually remain obscured by the diffusion, as they only indirectly affect the dMRI signal. The scattering formalism based on summing Feynman diagrams (Fig. 5) is key to solving the problem, since it allows us to consider the effect of realistic (i.e., structurally random, rather than periodic or fully-restrictive) along-axon geometries.”

R2-2: The relationship between axial diffusivity and bead-on-string axonal damage (which is a well-described outcome of TBI) is well known and well reported in the TBI and other neuroscience literature related to Wallerian degeneration (Examples include Song 2003; MacDonald 2007, but there are many). It has also been extended by characterization of axon diameter distribution (e.g. by DDE) and diffusion correlation imaging which offer improved specificity for beading (For example Komlosh 2018; Benjamini 2020). While the theoretical relationships presented are new and compelling, the idea that diffusion axial to beaded axons is decreased and that cross sectional variance increased is not new. If it is possible to show how the new model can be related to axial diffusivity or some of the other metrics, that might clarify the novelty of the work.

We agree that the general relationship between reduced axial diffusivity and bead-on-string axonal pathology is well documented in the literature on TBI and Wallerian degeneration (e.g., Song 2003; MacDonald 2007), and that other methods such as DDE and diffusion correlation imaging have aimed to improve specificity to axonal beading (e.g., Komlosh 2018; Benjamini 2020). But what distinguishes our work is that we do not empirically correlate diffusion-relevant parameters to histology; instead, we *derive* them analytically from first principles. Our main contribution lies not in targeting a specific disease model or dMRI imaging protocol, but in establishing a general analytical framework that exactly links axonal morphology to measurable diffusion parameters.

Specifically, we identify how two microstructural parameters, i.e., Γ_0 and $\langle 1/\alpha(x) \rangle$, govern the entire time-dependence of axial diffusivity $D(t)$ in the clinically relevant regime of long diffusion times. This establishes not just that $D(t)$ decreases, but *why* and *how* it does so as a function of geometry, offering a predictive closed-form forward model, rather than a correlative evaluation. This sets the stage for parameter estimation from dMRI, which we begin to demonstrate in our new *ex vivo* experiments, and which other groups will be able to build on in the future, using a plethora of dMRI techniques sensitive to the time-dependent diffusion. This has now been stated in the Discussion (page 9, lines 29-35).

“The present approach can be further incorporated into time-varying blocks of multiple diffusion encodings [6–8] or diffusion correlation imaging [9–14], thereby helping resolve contributions from multiple tissue compartments, and turning these techniques into deterministic methods to quantify axonal microgeometry and its changes in pathology, development and aging.”

R2-3: The analytic biophysical relationship between $D(t)$ and axon morphology as presented seems quite specific to axoplasm morphology only without considering myelin water diffusion or interstitial space. Although interstitial space is mentioned, it is dismissed as the same shape as the axonal space citing Nicholson, although this may not be the case depending on the (de)myelination, edema, gliosis, etc. Side note that EM preparation likely reduces or eliminates interstitial space. Why was the myelin not considered?

Indeed, the theoretical framework we present is focused on intra-axonal axial diffusivity, motivated by our overarching research philosophy of striving to deeply understand each phenomenon separately and build upon such understanding (see “Common response to all reviewers”). Each compartment (e.g., intra-axonal, myelin water, extra-axonal) is interesting in its own right, but the physics of diffusion is distinct in each of them, and modeling it requires distinct approaches and considering different relevant degrees of freedom. The exact relation between structurally disordered geometry and time-dependent diffusion has so far been unavailable for either of the compartments. This work symbolizes progress in the intra-axonal compartment, where the Fick-Jacobs equation is solved by representing it as a scattering problem.

Fortuitously, the intra-axonal space is the dominant contributor to the *axial* diffusivity, especially in coherent white matter bundles. The extra-axonal water has a lower volume fraction and a shorter T_2 relaxation time [15, 16], which lowers its relative contribution to the overall axial diffusivity. Furthermore, myelin water contributes minimally to the dMRI signal at clinically feasible echo times due to its short T_2 relaxation time [17]. This justifies the development, validation, and application of the present theory independent of other compartments.

We agree with the reviewers that myelin water and interstitial space may contribute in specific settings, especially while probing the diffusion transverse to fiber tracts, under the conditions of demyelination, edema, or gliosis. However, it turns out that the corresponding scattering approach in the extra-axonal compartment (transverse to axons, yielding the distinction between demyelination and axonal loss) requires quite a different treatment of the corresponding 2-dimensional Fick-Jacobs equation, and is far beyond the formalism developed in this work. Besides, the validation of any such theory with MC simulations presents its own set of challenges based on extra-axonal space shrinkage and the related challenges in segmentation. This approach is being actively pursued by our research group at the moment and will be presented separately.

We have changed “axial diffusivity” to “intra-axonal axial diffusivity” where it applies throughout the text. A con-

clusion of these considerations is now added to the Discussion (page 8, lines 83-115, page 9, lines 1-13).

R2-4: If I understand the description correctly, each axon segmentation or synthetic axon was handled individually, but their relative architecture was not addressed. Since the axons shown are far from parallel to each other and each would have relatively different orientation to any potential diffusion encoding direction, it is hard to understand how the diffusion and morphology parameters formulated for individual axons would be measurable within a voxel where relative axonal geometrics would become averaged. This seems necessary to address.

It is correct that our theoretical framework computes the diffusion time dependence $D_i(t)$ for each individual axon, based on its morphology. However, we explicitly address the voxel-level signal interpretation by incorporating ensemble averaging of these individual contributions. In particular, we define the voxel-averaged diffusion coefficient as: $D(t) = \sum_i w_i D_i(t)$, where w_i is the volume weight of the i th axon (see Main text, page 4, lines 77-83), assuming the voxel is dominated by a coherent population of axons with limited orientation dispersion. This weighted averaging mimics the contribution of each axon to the overall dMRI signal within a single imaging voxel. In practice, this approximation is valid in regions like the corpus callosum or cingulum, where orientation coherence is relatively high (We have now added a more comprehensive discussion on page 8, lines 83-115, page 9, lines 1-13.) We also rigorously included the effect of *axonal undulations* (Supplementary Information section: The harmonic undulation model) that is partly responsible for the observed orientation dispersion (refer to Page 4, Lines 37-45):

“*Undulations* [57] (wave-like variations of axon skeleton) cause a slower, $\sim 1/t$ tail [58] in $D(t)$, a sub-leading correction to Eq. (1).”

To further validate the connection between intra-axonal diffusivity and the observed dMRI signal along the fiber tract, we computed ensemble-averaged values of the geometric parameters $\langle 1/\alpha(x) \rangle$ and Γ_0 across thousands of segmented axons per specimen, and showed that the resulting predicted $D(t)$ curves (Fig. 3d-f) closely match the diffusion coefficients measured directly from *ex vivo* dMRI in the same anatomical region (see “Common response to all reviewers”). This agreement supports the idea that the proposed geometric descriptors can be meaningfully extended to ensemble-averaged metrics, and that their corresponding diffusion signatures are detectable at the voxel level.

R2-5: Minor comments

- Figures and sections are nice
- Writing is good.

- It’s a little hard to build an intuition for the variables and follow the math for the main theory (1-3) without a definition for CD which isn’t provided in the text. Is it just some variable called this or does it have a biophysical meaning in the model. It is later called one of the diffusion coordinates, but I think for a biophysical model to be well understood it is important to describe each variable, especially each of the 4 meaningful variables in the math. - Similarly, Γ_0 is a little non-intuitive way when defined (small q plateau of the power spectral density .. cross-sectional fluctuations). Overall it might be helpful to introduce the variables that are most consequential in a bit more detail.

We have revised the manuscript to explicitly define and contextualize the variables c_D and Γ_0 at the point of their introduction (see revised Theory section, page 2).

“In Eq. (3), $\Gamma_0 = \Gamma_\eta(q)|_{q \rightarrow 0} \dots$ The limit Γ_0 is a measure of axon shape heterogeneity at large spatial scales...”

“The theory (1)–(3) distills the myriad parameters ... and the amplitude c_D of its $t^{-1/2}$ power-law approach (i.e., the amplitude of the sub-diffusive correction to the growing mean-squared displacement $\ell^2(t)$).”

In the revised manuscript, we have expanded the discussion to more explicitly articulate the biophysical meaning of the model parameters (Main text, page 7, lines 7-222; Discussion, page 8, lines 98-110).

Response to Reviewer #3

Summary:

This study presents a novel theoretical relationship between two axonal morphology parameters and along-tract diffusion, potentially accessible through diffusion MRI (dMRI). The authors derive this relationship analytically using scattering theory and validate it through simulations based on axonal morphologies extracted from 3D histology. They then apply the framework to diffusion properties observed in rats with traumatic brain injury (TBI), showing that the derived morphological features and corresponding diffusion parameters can differentiate between injured and healthy tissue, highlighting their potential utility as biomarkers.

Strengths:

The key strength of the study lies in its combination of a novel, theoretically derived model and a robust validation demonstrating biomarker potential, within the model’s underlying assumptions. The authors present a strong theoretical framework that establishes a principled link between individual axon morphology and the diffusion properties along fiber tracts. The study uses high-resolution 3D histology and deep learning–based segmentation to accurately capture axonal morphology across large ensembles of axons. These realistic axon morphologies serve as the basis for Monte Carlo diffusion simulations, which offer a high-fidelity and biophysically grounded link between morphology and diffusion signal. The progressive validation strategy is fairly convincing: the authors first demonstrate strong correlation between morphological parameters estimated from simulated diffusion MRI and those extracted from histology, then show that axon populations can be discriminated between TBI and sham-operated rats, and finally demonstrate that this separation is preserved after ensemble-level volume averaging, mimicking an MRI-like scenario.

We thank the reviewers for highlighting the strengths of our work.

R3-1: Our main concern with the study is whether the proposed model can be translated into practical MRI experiments, given that it relies on histology-based simulations without validation using experimental diffusion MRI data. At the same time, the overall tone (see examples below) occasionally suggests a straightforward translation into practical diffusion MRI measurements without providing an example.

We have now added a full experimental dMRI section; please refer to “Common response to all reviewers”.

R3-2: Although the authors briefly acknowledge that additional signal contributions are present in water diffusion MRI, this limitation is not fully reflected in the clarity of the wording or the depth of the discussion. For example, the contribution of: the extra-cellular signal, orientation dispersion and crossing fibers to the proposed theoretical signal model are not addressed. For diffusion MR spectroscopy, a realistic discussion of scan time requirements is also missing.

Our theoretical model is formulated explicitly for intra-axonal diffusion, assuming geometrically disordered, impermeable axons, and does not incorporate signal contributions from extra-axonal space, myelin, or inter-compartmental exchange. This idealized assumption is clearly valid for our MC simulations (which are performed within segmented axons) and should be well applicable to the newly included *ex vivo* validation (in tightly packed, fixed tissue).

However, as mentioned in the original submission, and emphasized in the resubmission, in *in vivo* dMRI, the measured signal will include contributions from extra-axonal water, and this will numerically affect both c_D and D_∞ .

For that, we have now modified “axial diffusivity” to “intra-axonal axial diffusivity” where it applies throughout the text. As well, a conclusion of these considerations is now added to the Discussion (page 8, lines 83-115, page 9, lines 1-13).

R3-3: Another concern is the limited dataset. The authors progressively establish validation of the proposed model with Monte Carlo simulations for a small number of axons (100–150). In a second step, they apply the model to larger electron microscopy (EM) volumes totaling around 36,400 axons across five specimens to demonstrate sensitivity to group differences between TBI and sham-operated rats. While this stepwise approach provides convincing evidence and arguments to justify the progression between stages, the generalizability of the findings to actual MRI voxels remains somewhat limited. The total number of axons included in the study is smaller than what would be found in a single MRI voxel. Hence, it remains questionable whether the presented findings can generalize to the greater structural and pathological heterogeneity present in real-world MRI scenarios, especially considering that the analyzed axons represent a particular subset selected based on length thresholds. We suggest either (i) a more balanced acknowledgment of the practical challenges in making diffusion MRI measurements specific to the proposed parameters, or (ii) if feasible, some form of validation with experimental diffusion MRI data.

We offer an *exact relation* of time-dependent diffusion to the two morphological parameters of a random tube (such as an axon). To validate this exact theoretical relation, one does not need thousands of realizations of such a random tube — one needs to have a sufficient range of random tube shapes that represent generally random shapes. If the theory is wrong, it will already look wrong in many if not all of the 100 axons; if the theory is correct, a range of both artificial axons and EM-derived axons is enough. (Had our approach been approximate or “correlative” rather than analytical, reviewers’ concerns would have been valid.)

As the final step in our progressive model validation, we have now added a dedicated *ex vivo* time-dependent diffusion MRI experiment in two sham-operated and three TBI rats, along with a detailed discussion of confounding factors affecting the interpretability of the proposed parameters in real MRI acquisitions; please refer to “Common response to all reviewers”. The results demonstrate that the diffusion parameters c_D and D_∞ extracted from *ex vivo* MRI data fall within the predicted ranges derived independently from segmented 3d-EM data. Now the combination of (i) analytical modeling, (ii) MC validation, (iii) large-scale 3d-EM reconstructions, and (iv) experimental dMRI measurements, taken together, provides a robust, multi-level framework that supports the validity and applicability of our approach.

Regarding the concern about voxel-scale generalizability, we remark that our large field-of-view (FOV) 3d-EM datasets span nearly two animal MRI voxels. The 36,000 segmented axons are reconstructed with high fidelity from anatomically consistent white matter regions, across multiple animals, and including both healthy and disease conditions. These datasets constitute the largest 3d-EM resource of white matter, which we made publicly available. While the spatial resolution was intentionally traded for increased FOV, this introduced a higher rate of segmentation errors. The applied threshold in axonal length, keeping only sufficiently long axons, improved the estimation of geometric parameters—particularly Γ_0 , which captures short-range disorder—while preserving their anatomical representativeness.

R3-4: Examples Suggesting Straightforward Translation to MRI:

Page 4, line 109-111: “The validated theory opens up the way to massively speed up predictions of clinically feasible dMRI measurements and their change in pathology.”

The keyword in that sentence is *predictions*. Namely, we can predict $D(t)$ based on EM segmentations (obviously, *ex vivo*) in seconds rather than months or years of numerical calculations. This is not about clinical translation, but about building histology-informed hypotheses for subsequent clinical studies. “Clinically feasible” refers to the fact that subsequent $D(t)$ measurements would not require extreme scan times or gradient strengths.

We have now removed “Clinically feasible” from the sentence.

Page 6, line 4-7: “The validated theory opens up the way to massively speed up the predictions of dMRI measurements and their change in pathology based on 3d segmentations...”

R3-5: Page 6, line 42-44: “This representation mimics a dMRI measurement, demonstrating how MRI can capture TBI-related geometric changes in each voxel.”

We have now changed “dMRI measurement” or “axial diffusion” to “axial diffusion of the intra-axonal space” where it applies throughout the text:

Main text, page 5, lines 24-27):

“This representation mimics an axial diffusion tensor eigenvalue of the intra-axonal space within a coherent fiber tract, demonstrating how MRI can capture TBI-related geometric changes in each voxel.”

R3-6: Page 7, line 55-64: “An exact asymptotic solution of a key dynamical equation radically reduces the dimensionality of the problem: just two geometric parameters [...] enable a near-instantaneous prediction of a dMRI measurement from tens of thousands of axons.”

Conclusions and outlook, page 9, lines 55-57:

“...near-instantaneous prediction of an axial diffusion of the intra-axonal space from tens of thousands of axons”

R3-7: page 7, lines 20 -24: “Accessing the along-tract diffusion coefficient $D(t)$ corresponds to a clinically feasible, lowest-order $\sim q^2$ dMRI weighting, as opposed to very strong diffusion gradients (available on only a few custom-made scanners) required for mapping axon radii.” → But, in your simulations, you do not incorporate: a MRI-representative ensemble of axons; you do not incorporate extra-axonal signal; crossing fibres; fibres with different size (crosssectional and longitudinal).

Discussion, page 9, lines 18-19:

“Estimating the diffusion coefficient $D(t)$ of the intra-axonal space corresponds to clinically feasible dMRI weightings, as opposed to very strong diffusion gradients...”

Additionally, with the added DTI experiment and Fig. 4, we strengthened our discussion about the approximate applicability of along-tract axial diffusivity $D(t)$ derived from the time-dependent DTI in major fiber tracts.

Specific questions/suggestions/comments:

R3-8: page 4, lines 68-72: “[...]MC-simulated along-axon diffusion coefficients $D_i(t)$ in the i -th axon were volume-weighted (corresponding to spins’ contributions to the dMRI signal) to produce the ensemble-averaged $D(t) = \sum_i w_i D_i(t)$ [...]” Could you introduce the assumptions regarding fiber configuration and signal compartments? In particular, Is it also valid if extracellular signal is present and for crossing fibres (if so please explain why)? Where was this relation derived? Does it come from Lee et al., 2020 (Eq. 10a)?

The derived relation for the ensemble-averaged diffusion coefficient, $D(t) = \sum_i w_i D_i(t)$, assumes that the signal comes from intra-axonal water, with $D_i(t)$ representing the time-dependent axial diffusivity within the i th axon and w_i its volume fraction. This expression follows directly from the additivity of signal contributions that can be approximated by the cumulant expansion $S(b, t) = e^{-bD(t) + \frac{1}{6}b^2D^2(t)K(t) + \mathcal{O}(b^3)}$ (Lee et al., 2020 [Eq. 10a]), under the assumption of no exchange between compartments and axons being spatially aligned (no fiber crossings). This assumption is valid in tightly packed white matter regions, where permeability is low and intra-axonal compartments dominate the long-time axial signal.

R3-9: Are the sketched axons in Fig. 2a true to scale? And why does the diameter of the synthetic axon appear larger than the one of the segmented axons? Why did you choose $\alpha - 1$ on the y -axis?

Thank you for pointing this out.

-In Fig. 2a, the TBI axon was scaled with 1:1.15 length-to-diameter to ensure cross-sectional variations are visible. We have now fixed this in the revised figure so that all axons are rendered true to scale (see Fig. 2a on Page 3).

-The cross-sectional diameters of synthetic axons are larger than EM axons: “we fixed $A_0 = \pi \cdot (0.5)^2 \mu\text{m}^2$ ” in *Methods*; Synthetic axon generation. This allowed us to apply stronger cross-sectional variations to the synthetic axons and avoid forming small bottlenecks for MC simulations. Stronger variations allow us to test our theory covering a larger range of random axonal shapes, making its validation more convincing.

-In Fig. 2a, the y -axis represents the normalized cross-sectional fluctuations $\delta\alpha = \alpha - 1$. Plotting $\delta\alpha$ instead of α directly centers the baseline around zero, making geometric fluctuations more visually interpretable.

R3-10: Fig. 2d: how does the Gaussian kernel length in the caption relate to the sketched axons ? It might also be helpful to highlight that the right-most color-bar is not another axon.

The coarse-graining of the axon segment is color-coded based on the increasing diffusion time t_i , where the standard deviation of the Gaussian kernel is calculated as $\ell(t_i)/\sqrt{2}$. We have now modified the caption of Fig. 2d to the following:

“(d) Coarse-graining over the increasing diffusion length ... axon segment is Gaussian-filtered with the standard deviation $\ell(t_i)/\sqrt{2}$ for $\ell(t_i) = 0, 5, 10, 20 \mu\text{m}$. The coarse-graining of the axon segment along its length is color-coded for increasing diffusion times t_i according to the color bar.”

R3-11: Synthetic axons in Fig. 2: Why were those numbers ($L=500 \mu\text{m}$ and $N=50$) chosen? How do the results change as a function of L ?

Increasing L improves the spectral resolution for estimating $\Gamma_\eta(q)$ behavior at low- q , but at the same time increases the computational burden of MC simulations both on memory and number of random walkers. $L = 500 \mu\text{m}$ is long enough to ensure an accurate measure of $\Gamma_\eta(q)|_{q \rightarrow 0}$ yet computationally feasible for MC simulation. For our purposes, the gain becomes marginal by increasing L beyond $500 \mu\text{m}$. The number of synthetic axons $N = 50$ ensures an adequate statistical power when plotting geometry vs diffusion-related parameters (see also our discussion on the

validation of an exact theory above).

R3-12: The color coding in Fig. 2e–f may benefit from improved contrast to support clearer differentiation.

We improved the contrast in Fig. 2e–f colormaps on Page 3.

R3-13: The definition of error bars in Figure 2, 3 and S2–4 could be clarified. Are these errors standard errors arising from linear fits of diffusion parameters and polynomial fit of Γ_0 ? If so, errors could be shown in $D(t)/D(\infty)$ over $1/\sqrt{t}$ graphs as well.

- Figure 2e: Horizontal error bars are the standard deviations of D_∞ , estimated over a range of plausible lower-bound diffusion times $t_0 \in [3, 10]$ ms to restrict the fit to the asymptotic regime where the power-law scaling holds. This variability dominates over individual fit uncertainty.
- Figure 2f: Horizontal error bars for c_D follow the same procedure as in Fig. 2e. Vertical error bars are the standard deviations in estimating Γ_0 over a range of fractions $\beta \in [0.92, 0.97]$ of the total variance of the cross-sectional fluctuations (Eq. S1). This variability dominates over individual fit uncertainty.
- Figure 3e: Error bars are the weighted standard deviations in estimating D_∞ and c_D from the distributions of diffusion parameters of individual axons in (c).
- Figure 3g: Error bars correspond to propagated standard deviations of $\langle 1/\alpha \rangle$ and Γ_0 , based on errors in measuring the volume-weighted D_∞ and c_D in (e).
- Following the reviewer’s suggestion, we now include error bars in the plots of $D(t)/D_0$ vs $1/\sqrt{t}$ in Fig. 3f and Supplementary Figs. S2f–S4f using the same procedure.

We collected this information and added a new section, Estimation of uncertainties and error bars, in *Methods* on page 19. We also modified figure captions by referring readers to *Methods* where it applies.

R3-14: How do the synthetic axons help “reducing errors in estimating Γ_0 ” for EM-based axons? Was the information from synthetic axons directly used for fitting Γ_0 of EM-based axons? See page 4, lines 67–72.

Synthetic axons were not used in fitting Γ_0 for EM axons; rather, they served as a validation tool. Since increasing L improves the spectral resolution for estimating $\Gamma_\eta(q)|_{q \rightarrow 0}$, we generated long synthetic axons, $L = 500 \mu\text{m}$, to confirm that the same polynomial fitting procedure applied to EM axons can reliably recover Γ_0 of long synthetic axons.

R3-15: The authors suggest an absence of bias in Figure 2f. However, there appears to be a slight systematic deviation for synthetic axons, which could also be present but obscured by the high density of data points for EM-segmented axons. Could you provide a plot in which points have less overlap? Could you comment on what might underlie the slight bias?

The deviation from the identity line becomes more apparent with increasing cross-sectional variation $\text{var}[\alpha(x)]$. We attribute these deviations to corrections to the FJ equation (4) when the assumption of full decoupling between fast transverse and slow longitudinal dynamics begins to break down. This has been stated in the Main text on page 4, lines 108–113:

“Figure 3f also indicates that as the cross-sectional variation $\text{var}[\alpha(x)]$ increases, the deviations from the identity line become more pronounced, which could be attributed to corrections to FJ equation (4), when the “fast” transverse and “slow” longitudinal dynamics are not fully decoupled.”

We attached enlarged panels of Fig. 2f to the response letter.

R3-16 Page 4, line 109: “The validated theory opens up the way to massively speed up predictions of clinically feasible dMRI measurements and their change in pathology: [...]” As I understand it, the validated theory does not speed up clinically feasible dMRI measurements, but might potentially enable them in the first place.

FIG. 2. ... (f) Predicted c_D , Eq. (3), validated against its MC counterpart estimated from Eq. (1), for individual synthetic and TBI axons (colors as in (a)). Coefficient c_D is larger for axons with greater cross-sectional variations. Error bars reflect errors in estimating c_D from MC-simulated $D(t)$ (horizontal) and estimating the plateau Γ_0 from $\Gamma_\eta(q)$ (vertical), as shown by dashed lines in panel (a).

This is a correct statement: As in R3-4, the validated theory massively speeds up the *prediction* of axial $D(t)$ from the intra-axonal space by eliminating the need for MC simulations. This is essential for model-building and creating educated hypotheses based on histology for future clinical studies in a wide range of neurological and neurodegenerative disorders

R3-17 Fig. 3f: Is the observed difference in $D(t)$ statistically relevant? How do the variations in "e" propagate into "f"?

Due to the small number of animals (3 TBI, 2 Sham-operated), we refrain from formal statistical hypothesis testing. The observed trends in both D_∞ and c_D consistently distinguish the two groups, as shown in the feature space (Fig. 3e-f), and align with geometric predictions derived from histology. These results support the potential sensitivity of the proposed metrics. Please also refer to R3-13 and the newly added section, Estimation of uncertainties and error bars, in *Methods* on page 19, where we show how variations in Fig. 3e propagate into Fig. 3f.

R3-18 Page 7, line 35-39: "For water dMRI, we expect the along-axon extra-axonal space contribution to be similar given that its geometric profile mirrors that of intra-axonal space [...]." The statement may overstate the similarity between intra- and extra-axonal contributions. Could you elaborate on why this is your expectation?

The similarity holds under the assumption of tightly packed coherent bundles, where the extra-axonal geometry closely mirrors the axonal layout [18]. This is further limited to regions with negligible edema or demyelination. Of course, this is an approximation; the ultimate test is the experiment, and we are encouraged by the newly presented dMRI results. These considerations are now added more comprehensively to the Discussion, page 8, lines 83-115, page 9, lines 1-13.

R3-19 Page 7, line 30-34: The authors introduce the paragraph with the clinical feasibility in terms of "lowest-order q^2 weighting". How do the practical methods mentioned in this paragraph fit into other aspects of clinical feasibility, e.g., scan times, voxel size in diffusion MR spectroscopy?

The added experimental section shows that the predicted geometric parameters, Γ_0 and $\langle 1/\alpha(x) \rangle$ can indeed be reflected in experimentally measurable diffusion parameters, c_D and D_∞ , respectively. The close agreement between histologically predicted and experimentally observed values provides strong initial support for the practical measurability of these descriptors. See also the answer to R1-2.

FIG. S9. Axonal length and volume. Distributions of axonal length l and volume V for all myelinated axons included in this study. Axons shorter than $70, \mu\text{m}$ in the corpus callosum and $40, \mu\text{m}$ in the cingulum were excluded. While axonal lengths remain comparable between groups, the reduced axonal volumes observed in the TBI group reflect a decrease in cross-sectional diameters. This is also shown in Fig. S5 (top row) across brain regions: Cg-ipsi ($N = 3,999$), Cg-contra ($N = 4,580$), CC-ipsi ($N = 2,703$), and CC-contra ($N = 4,080$).

R3-20 Page 6, line 57-59: The authors attribute the stronger effect size in volume-weighted diffusion metrics to axon radius differences. Possibly, imperfect segmentation could confound this observation. Showing not only distributions of axon radii but also including axon length and volume could support the authors’ attribution of volume-change mainly to axons radii decrease more unambiguously.

From page 15, lines 20-25: “In our analysis, we only included axons that were longer than $70 \mu\text{m}$ in the corpus callosum and $40 \mu\text{m}$ in the cingulum. We further excluded axons with protrusion causing bifurcation in the axonal skeleton and axons with narrow necks with a cross-sectional area smaller than nine voxels for MC simulations.”

Applying a threshold over the length and excluding axons with protrusion, we circumvented the confound of segmentation error and axonal length variability. We now include distributions of axon length and volume in a **new Supplementary Fig. S12**, showing that axon radius accounts for the majority of volume change.

R3-21 Page 7 line 3-5: “The repercussions of the laterally induced brain injury affected both the cingulum and corpus callosum, with a pronounced effect on the cingulum.” It’s unclear to what repercussion in the corpus callosum the authors refer.

The effect refers to changes in cross-sectional variance, $\text{var } \alpha$, which were less pronounced in the corpus callosum than in the cingulum. We now have modified the text on page 8, lines 41-45, as follows:

“The laterally induced brain injury leads to **increased cross-sectional variance $\text{var}[\alpha(x)]$** in both the cingulum and corpus callosum, with a more pronounced effect observed in the cingulum.”

R3-22 Discussion: the authors don’t discuss the impact of axon selection by length on the observed results

Axons are supposed to be longer than a threshold that we have a better measure of Γ_0 as accessing low-q requires longer axons. Axons were thresholded by length to ensure accurate estimation of Γ_0 , which requires access to low-q modes. This may bias toward more coherent axons, but it is necessary for reliable measurement.

R3-23 What about inter-individual differences as a confounder? E.g., Fig. S3a-c suggest great similarities between Sham#25 and TBI#28 than within groups.

The animals used in this study are of the same age and gender, and both sham-operation and TBI procedures were applied using consistent protocols. The SBEM datasets were acquired from anatomically matched regions in the white matter for both sham-operated and TBI animals. Given the limited sample size, we acknowledge that it is difficult to draw firm conclusions regarding the role of inter-individual variability. Further studies with larger cohorts would be needed to systematically address this potential confounder. We remark that from the histology perspective, this is nevertheless one of the largest 3d-EM datasets of white matter available to date.

R3-24 Page 6, line 36: “[...] five distinct voxels in five animals with aligned impermeable myelinated axons.” What is meant with voxels? How do they relate to MR voxels?

Here, “voxels” refer to virtual voxels constructed from segmented EM regions. These are not MRI voxels but represent analogous subvolumes used for ensemble averaging. We now have modified the text as follows (page 7, line 17):

“...originating from five distinct voxel-like subvolumes in five animals ...”

R3-25 Page 6, lines 40 - 42: Plugging the cD and D_∞ values into Eq. (1), we predict the associated $D(t)$ for these five voxels as a function of $1/\sqrt{t}$ in Fig. 3d. ” → Shouldn’t this be Fig. 3f?

Thank you for noticing this. We corrected “...Fig. 3d” to “...Fig. 3f”.

R3-26 Figure S1: The fitting looks very unstable. Can you motivate why a second-order polynomial can be used for the fitting and how the constraints are motivated? How does a potential error in Γ_0 propagate into the group difference analysis in Fig. 3 etc.

The second-order polynomial $\Gamma_\eta(q) \approx \Gamma_0 + a_2q^2$ captures the low- q behavior of $\Gamma_\eta(q)$, which is even in q . The empirical parameter β in Eq. S1 sets the upper wavevector cutoff q_{\max} on the cumulative power such that the integral $\int_0^{q_{\max}} \frac{dq}{2\pi} \Gamma_\eta(q)$ contains a fixed fraction of the total variance. Note that the contribution of $\Gamma_\eta(q)$ to the integral saturates at high q_{\max} , marking the onset of a plateau. Plotting Γ_η on a linear scale in Fig. S1 better demonstrates this transition.

R3-28 As suggested by the reviewer, we further discussed the error propagation in estimating Γ_0 as a separate section; please refer to R3-13.

R3-29 Page 6, line 25: The link to Eq. (S22) is set wrongly.

Thank you for noticing this. Corrected in the revised manuscript.

R3-30 Figure S5 caption: “The histograms demonstrate the decrease in the axon radius caused by TBI on the ipsilateral side.” This is only true for the cingulum.

Indeed, all histograms demonstrate a decrease in axon radius caused by TBI, as indicated by the negative effect sizes. While the magnitude of the changes may be modest, in regions other than Cg ipsilateral, the directionality is consistent across all brain regions.

R3-31 The effect size is defined in terms of the median absolute deviation, but the results section refers to it as “standard deviations” See, e.g., page 6, line 15; and page 6, line 23.

Thank you for noticing this: we have now modified the text, changing standard deviation to median absolute deviation on page 6, line 19, and page 7, line 5.

R3-32 What is the motivation for the relative sample sizes (57 TBI vs. 43 sham axons)? See page 4, lines 62–67.

The relative sample sizes (57 TBI vs. 43 sham axons) simply followed the 2:3 ratio of sham to TBI animals in our dataset.

R3-33 Page 7, lines 23 - 24: “Moreover, in Supplementary Eq. (S7), we show that [...]” → It is not linked to Eq. S7.

Thank you for noticing this. Corrected in the revised manuscript.

R3-34 Equation (22): Please provide intuitive information about “C”

In Eq. 22, the constant C is the amplitude of the power-law scaling of $\Gamma_\eta(q)$ at small q . We added the following to page 18, line 10:

“...consider the small-wavevector fluctuations of $\ln \alpha(x)$:

$$\Gamma_\eta(q) \simeq C|q|^p, \quad q \rightarrow 0, \quad (2)$$

where the constant C is the amplitude of the power-law scaling. We can call random tubes...”

R3-35 Reviewer #3 (Remarks on code availability):

Code Availability:

The authors share the code for axon segmentation and Monte Carlo simulations, providing access to the main building blocks of their analysis pipeline. However, the code for reproducing the specific results presented in the manuscript is not publicly available. We have not attempted to test the provided code.

The data and codes used for generating the results presented in the manuscript are now publicly available. Please refer to the Data and Code availability sections on pages 9-10.

Response to Reviewer #4:

We appreciate the time and effort invested in reviewing our manuscript, especially the involvement of Early Career Researchers as part of the co-review process.

-
- [1] E. Fieremans, L. M. Burcaw, H. H. Lee, G. Lemberskiy, J. Veraart, and D. S. Novikov, *NeuroImage* **129**, 414 (2016).
 - [2] H.-H. Lee, A. Papaioannou, S.-L. Kim, D. S. Novikov, and E. Fieremans, *Communications Biology* **3**, 354 (2020).
 - [3] A. Arbabi, J. Kai, A. R. Khan, and C. A. Baron, *Magnetic Resonance in Medicine* **83**, 2197 (2020), _eprint: <https://onlinelibrary.wiley.com/doi/pdf/10.1002/mrm.28083>.
 - [4] E. T. Tan, R. Y. Shih, J. Mitra, T. Sprenger, Y. Hua, C. Bhushan, M. A. Bernstein, J. A. McNab, J. K. DeMarco, V. B. Ho, and T. K. F. Foo, *Magnetic Resonance in Medicine* **84**, 950 (2020), _eprint: <https://onlinelibrary.wiley.com/doi/pdf/10.1002/mrm.28180>.
 - [5] J. Valette, C. Ligneul, C. Marchadour, C. Najac, and M. Palombo, *Frontiers in Neuroscience* **12**, 10.3389/fnins.2018.00002 (2018), publisher: Frontiers.
 - [6] N. Shemesh and Y. Cohen, *Magnetic Resonance in Medicine* **65**, 1216 (2011).
 - [7] S. N. Jespersen, H. Lundell, C. K. Sönderby, and T. B. Dyrby, *NMR in Biomedicine* **26**, 1647 (2013).
 - [8] N. Shemesh, S. N. Jespersen, D. C. Alexander, Y. Cohen, I. Drobnjac, T. B. Dyrby, J. Finterbusch, M. A. Koch, T. Kuder, F. Laun, M. Lawrenz, H. Lundell, P. P. Mitra, M. Nilsson, E. Ozarslan, D. Topgaard, and C.-F. Westin, *Magnetic Resonance in Medicine* **75**, 82 (2015).
 - [9] P. T. Callaghan and I. Furó, *The Journal of Chemical Physics* **120**, 4032 (2004).
 - [10] M. E. Komlosh, F. Horkay, R. Z. Freidlin, U. Nevo, Y. Assaf, and P. J. Basser, *Journal of Magnetic Resonance* **189**, 38 (2007).
 - [11] E. Ozarslan, *Journal of Magnetic Resonance* **199**, 56 (2009).
 - [12] M. E. Komlosh, E. Özarslan, M. J. Lizak, F. Horkay, V. Schram, N. Shemesh, Y. Cohen, and P. J. Basser, *Journal of Magnetic Resonance* **208**, 128 (2011).
 - [13] J. P. de Almeida Martins and D. Topgaard, *Physical Review Letters* **116**, 087601 (2016).
 - [14] D. Benjamini, E. B. Hutchinson, M. E. Komlosh, C. J. Comrie, S. C. Schwerin, G. Zhang, C. Pierpaoli, and P. J. Basser, *NeuroImage* **221**, 117195 (2020).
 - [15] J. Veraart, D. S. Novikov, and E. Fieremans, *NeuroImage* **182**, 360 (2018).
 - [16] C. M. Tax, E. Kleban, M. Chamberland, M. Baraković, U. Rudrapatna, and D. K. Jones, *NeuroImage* **236**, 117967 (2021).
 - [17] R. D. Dortch, K. D. Harkins, M. R. Juttukonda, J. C. Gore, and M. D. Does, *Magnetic Resonance in Medicine* **70**, 1450 (2013), _eprint: <https://onlinelibrary.wiley.com/doi/pdf/10.1002/mrm.24571>.
 - [18] E. Syková and C. Nicholson, *Physiol Rev* **88**, 1277 (2008).

Scattering approach to diffusion quantifies axonal damage in brain injury

Ali Abdollahzadeh,^{1,2,*} Ricardo Coronado-Leija,¹ Hong-Hsi Lee,³
Alejandra Sierra,² Els Fieremans,¹ and Dmitry S. Novikov^{1,*}

¹*Center for Biomedical Imaging, Department of Radiology,
New York University School of Medicine, New York, NY, USA*

²*A.I. Virtanen Institute for Molecular Sciences, University of Eastern Finland, Kuopio, Finland*

³*Athinoula A. Martinos Center for Biomedical Imaging, Department of Radiology,
Massachusetts General Hospital, Harvard Medical School, Boston, MA, USA*

* Corresponding authors:
ali.abdollahzadeh@uef.fi, dmitry.novikov@nyulangone.org

REVIEWER COMMENTS

NCOMMS-25-03290A-Z

Response to Reviewer #1

The authors have satisfied my comments with their revisions.

We thank the reviewer for their thoughtful review and constructive feedback. We are glad the revisions addressed your comments and appreciate your time and effort in evaluating our work.

Response to Reviewer #2:

The revised manuscript Scattering approach to diffusion quantifies axonal damage in brain injury describes an analytic biophysical relationship between axon cross sectional variability and water diffusion providing a bridge between this specific cellular feature and the physical process of water diffusion that is measurable by MRI. The model could be a foundation for more specific non-invasive mapping of axonal features in disease states (e.g. TBI) in the future. The model is validated using Monte Carlo simulations with SBEM data and the additional ex vivo diffusion MRI experiments were performed with rat brain tissue after mild TBI to demonstrate how the model may provide axon specific metrics in disease states. The main contribution of the work is the theoretical development of a model for axonal geometry and the simulation and experimental work supporting the model is appropriate. The revised writing is much improved, especially the goals of the work are more clear and the variables of the model are defined in a way that builds an intuitive understanding. The impact seems modest. While it is not clear that developing this model will have practical implications that improve what already exists for detection of axonal pathology after TBI or in other disease states, the development of the theory is creative and nicely done and the direct analytic bridging of axonal microgeometry and diffusion is compelling. The methodology is sound. The manuscript and figures are appropriate and I suggest no revision.

Thank you for the thoughtful and encouraging evaluation of our work. We appreciate your thoughtful assessment of our theoretical framework linking axonal microgeometry to measurable diffusion signals. While we acknowledge that broader translational impact may require future studies, we are glad that you find the study creative, analytically grounded, and methodologically sound. We appreciate your time and effort in reviewing our work.

Response to Reviewer #3

We appreciate that the authors have addressed most of our comments and the additional *ex vivo* experiment complemented and increased the relevance of the paper. In total, there is significant innovation and evidence for the important claims in the paper, and therefore we recommend publication. We would also like to agree with the initial statement of the authors where they highlighted the strength of their paper and emphasize that questioning those contributions are and were not subject of our comments.

We thank the reviewer for their thoughtful review and constructive remarks. We are glad the revisions addressed most of the comments, and we appreciate your time and effort in evaluating our work.

However, we would recommend that the limitations of the methods are clearly stated in the discussion and that some of the current statements are toned down accordingly. Please consider the following comments:

R3-1: It seems that the authors agree that it is unclear for *in vivo* measurements whether signal will be mainly coming from the intra-cellular compartment and thus the estimated axial diffusivity can be interpreted in terms of the proposed modelled signal. This is particularly clear from the answer to R3-2. However, this is not adequately reflected in the discussion. We suggest that parts of the answer go into the discussion, so that not only the reviewers but all readers can benefit from this clarification. Here are the relevant passages: “Our theoretical model is formulated explicitly for intra-axonal diffusion, assuming geometrically disordered, impermeable axons, and does not incorporate signal contributions from extra-axonal space, myelin, or inter-compartmental exchange. [...] in *in vivo* dMRI, the measured signal will include contributions from extra-axonal water, and this will numerically affect both cD and $D\infty$. [...]”

We have now modified the Discussion section, explicitly stating the assumptions of our theoretical model and the multi-compartment nature of a measured dMRI signal (page 8, lines 108-112, page 9, lines 1-35).

“For water dMRI, our theory is formulated explicitly for intra-axonal diffusion, assuming geometrically disordered, impermeable axons, and does not incorporate signal contributions from extra-axonal space, myelin, or inter-compartmental exchange ...

Measuring the along-tract diffusion tensor eigenvalue (axial diffusivity) via the lowest-order in diffusion weighting $b \sim q^2$, as in Fig. 4, is much more straightforward, yet it is confounded by the extra-axonal water contribution. This confound would be minimal for highly aligned axonal tracts, with tightly packed axons, where the extra-axonal water fraction is the smallest [78-80], and is further suppressed by the relatively faster extra-axonal T_2 relaxation [80-82]. Furthermore, the extra-axonal contribution is qualitatively and quantitatively similar [83], given that its geometric profile mirrors that of intra-axonal space ...

Increasing the fiber orientation dispersion would increase the confounding effects. Injury-induced extracellular processes, such as inflammation or glial proliferation, can further modulate the extra-axonal contribution.

Remarkably, our *ex vivo* dMRI experiment shows that the along-tract DTI measurement already captures the effect of mild TBI, Fig.4 ... that the intra-axonal contribution to the overall along-tract DTI eigenvalue dominates in the highly aligned tracts, at least in the *ex vivo* setting.”

R3-2: In this context, the following statements in the paper need clarification:

- Page 8, lines 60-64: “While it has been observed that the long-time asymptote $D\infty$ and the amplitude cD are qualitatively affected by axonal beadings [12, 46, 51], their exact relations with tissue microgeometry have remained unknown. Our scattering approach solves this fundamental problem, [...]”

Assuming that there is extra-cellular signal contribution in the estimated diffusion parameters in *in vivo* application, it is questionable that the employed model can explain the “exact relations with tissue microgeometry”.

The exactness of relations with tissue microgeometry refers to the solution of the FJ equation in the intra-axonal space.

We added “for the intra-axonal space” to be specific.

R3-3: - Page 8; lines 103-106: “Remarkably, our dMRI experiment shows that even the lowest-order in diffusion weighting $b \sim q^2$, the along-tract DTI, already captures the effect of mild TBI, Fig. 4.” For *ex-vivo* MRI only.

Please refer to R3-1 for the modified Discussion section.

R3-4: - Page 9, lines 59 – 63: “ These two relevant parameters are sensitive to the variation of the cross-sectional area and the statistics of bead positions, opening a non-invasive window into axon shape alterations three orders of magnitude below the resolution of clinical MRI.”

Clinical MRI is typically applied *in vivo*, but *in vivo* the validity of the employed model is unclear due to the unknown extra-cellular signal contribution.

We believe that our solution provides a crucial building block for non-invasive, model-based quantification of axon (as well as potentially dendrite and glia) shapes, given that it can be used either directly with diffusion-weighted spectroscopy (see R3-7 below) or through multi-compartment modeling. To put things in perspective, we removed the word “clinical”, since clinical protocols are quite short, whereas we are confident this building block can and will be used in a research setting (both in animal and in human MRI). In fact, our team is pursuing such a research program at the moment.

R3-5: - Page 9, lines 69 – 76: “It can be used to detect previously established μm -scale changes that occur not only in axons but also in the morphology of dendrites during aging [97, 98], and pathologies such as stroke [99], Alzheimer’s [100] and Parkinson’s [101, 102] diseases, and amyotrophic lateral sclerosis [103, 104], with an overarching aim of turning MRI into a non-invasive *in vivo* tissue microscope.”

Again, the validity of the established relationship for *in vivo* application is unknown due to the unknown extra-cellular signal contribution.

Our solution is a fundamental solution enabling a key building block for different *in vivo* approaches; see also the response to R3-4 above and R3-7 below. We slightly edited the text as follows (Page 9, lines 97 – 98):

“This key modeling building block can be used to detect previously established μm -scale changes that occur not only in axons but also in the morphology of dendrites during aging [97, 98], and pathologies such as stroke [99], Alzheimer’s [100] and Parkinson’s [101, 102] diseases, and amyotrophic lateral sclerosis [103, 104], with an overarching aim of turning MRI into a non-invasive *in vivo* tissue microscope.”

R3-6: 2. In the discussion of the *ex vivo* data, there is an interpretation on page 8 (lines 108 – 114) which could be further toned down: “We believe this is because the intra-axonal contribution to the overall along-tract DTI eigenvalue dominates in the highly aligned tracts, while the extra-axonal contribution is also qualitatively and quantitatively similar [78], given that its geometric profile mirrors that of intra-axonal space.”

Also other effects in the extra-cellular signal unrelated to the “geometric profile mirroring that of intra-axonal space” could also lead to a reduction of the diffusion coefficient such as inflammatory processes and thereby potentially also produce a similar behaviour as that one observed experimentally.

We agree with the reviewer that extracellular processes unrelated to geometric mirroring, such as inflammation or glial proliferation, can indeed modulate diffusion behavior. This insight has now been added to the Discussion (page 9, lines 23 – 26):

“Injury-induced extracellular processes, such as inflammation or glial proliferation, can further modulate the extra-axonal contribution.”

R3-7: 3. The mentioning of diffusion-weighted spectroscopy as a useful method for this application, is still unclear to us. For example on page 8, lines 83-87, the authors state: “The most direct and specific access to the time dependent diffusion within the intra-cellular space can be provided by diffusion-weighted spectroscopy [47, 48, 71–74] of intracellular metabolites, such as N-acetylaspartate (NAA) for the axons. The same NAA-based approach can be used to probe the cross-sectional variations of the dendritic processes in the gray matter. [...]”

If we understand the authors correctly, they claim that their model can be applied to the time dependent diffusion data acquired with those NAA-based approaches. But, later in the discussion, the authors highlight that their model is only valid for highly aligned fibres (page 9, lines 3-9). Clearly, in the structures investigated (*in vivo*) by the papers cited above, the fibres were not highly aligned and in gray matter this assumption is also not valid. Could you please clarify this contradiction?

There is a considerable difference between water (which is everywhere, requiring aligned fibers), and intracellular metabolites, where no orientation dispersion requirement is necessary.

The text explicitly states (we edited this sentence for clarity in the current resubmission): “The trace of the time-dependent diffusion tensor $D_{ij}(t)$ for the corresponding intracellular metabolite at long t would yield the time-

dependent diffusion coefficient $D(t)$, Eq. (1), along the structurally disordered neuronal or glial cell processes, independent of their orientational dispersion.”

In other words, for a signal solely coming from a “stick” compartment of a single kind (i.e., either dendrites in gray matter, or axons in white matter, or glia processes), the orientations of these “sticks” are irrelevant and their alignment is not necessary. This is because the diffusion-weighted signal in a unit direction \mathbf{g} from a collection of “sticks” pointing in all possible unit directions \mathbf{n} with the orientation dispersion $\mathcal{P}(\mathbf{n})$,

$$S(\mathbf{g}) = \int_{|\mathbf{n}|=1} d\mathbf{n} \mathcal{P}(\mathbf{n}) e^{-bD(t)(\mathbf{n}\cdot\mathbf{g})^2} = 1 - \sum_{i,j=1}^3 b g_i g_j D_{ij}(t) + \mathcal{O}(b^2),$$

in the DTI approximation $\sim b$ yields the diffusion tensor

$$D_{ij}(t) = D(t) \int d\mathbf{n} \mathcal{P}(\mathbf{n}) n_i n_j, \quad \text{tr } D_{ij}(t) = D(t),$$

where $D(t)$ is the along-stick (i.e., along-axon or along-dendrite) diffusion coefficient for which our theory has been developed. Remarkably, taking the trace, using $\text{tr}(n_i n_j) = \sum_i n_i^2 = 1$, cancels out the effect of the orientation dispersion due to its normalization $\int d\mathbf{n} \mathcal{P}(\mathbf{n}) \equiv 1$ onto unit probability. (Here we assumed the time t to be sufficiently long so as the transverse-to-stick motion is coarse-grained to yield the corresponding 1-dimensional Fick-Jacobs description.)

This consideration makes our approach very lucrative for diffusion-weighted spectroscopy. We would like to give it full credence in the upcoming line of work being actively pursued by our team.

R3-8: 4. In the discussion the authors sound very optimistic about the in vivo applicability of along-track $D(t)$ (Page 9, lines 18-22) for the intra-axonal signal and put it above high- b -value measurements in terms of its practicability: “Estimating the diffusion coefficient $D(t)$ of the intra axonal space corresponds to clinically feasible dMRI weightings, as opposed to very strong diffusion gradients (available on only a few custom-made scanners) required for mapping axon radii [87].”

The time-dependent approach is of course of higher practicability than the high- b -value approaches, but as long as the role of the extra-cellular signal contribution for in vivo measurements is not clarified, it is unclear whether the proposed model is superior to the ref approach.

We now understand the source of confusion: this sentence followed the along-track DTI estimation description, and implied it, which of course would be confounded by the extra-axonal water. We now merged that part with the previous paragraph that is now fully devoted to along-track DTI results.

The sentence in question now starts a new paragraph meant to use our $D(t)$ in settings where contributions from different compartments are assumed to be distinguished. Therefore, we added “with time-dependent multi-compartment modeling or spectroscopy” to imply that the purely intra-axonal contribution has been identified. Note that the time-dependent multi-compartment modeling requires moderate (available on modern clinical systems) but not super-high (Connectom or Magnus-level) gradients.

Response to Reviewer #4:

We appreciate the time and effort invested in reviewing our manuscript, especially the involvement of Early Career Researchers as part of the co-review process.